# Segmenting Text and Learning Their Rewards for Improved RLHF in Language Model

**Yueqin Yin**[*]                                           *yueqin.yin@utexas.edu*
*The University of Texas at Austin, Microsoft*

**Shentao Yang**[*]                                   *shentao.yang@mccombs.utexas.edu*
*The University of Texas at Austin*

**Yujia Xie**[†]                                           *yujiaxie@microsoft.com*
*Microsoft*

**Ziyi Yang**[†]                                           *ziyiyang@microsoft.com*
*Microsoft*

**Yuting Sun**                                           *yutingsun@microsoft.com*
*Microsoft*

**Hany Awadalla**                                           *hanyh@microsoft.com*
*Microsoft*

**Weizhu Chen**                                           *wzchen@microsoft.com*
*Microsoft*

**Mingyuan Zhou**[†]                                   *mingyuan.zhou@mccombs.utexas.edu*
*The University of Texas at Austin*

**Reviewed on OpenReview:** *https://openreview.net/forum?id=YhLlqDOUNi*

## Abstract

Reinforcement learning from human feedback (RLHF) has been widely adopted to align language models (LMs) with human preference. Previous RLHF works typically take a bandit formulation, which, though intuitive, ignores the sequential nature of LM generation and can suffer from the sparse reward issue. While recent works propose dense token-level RLHF, treating each token as an action may be oversubtle to proper reward assignment. In this paper, we seek to get the best of both by training and utilizing a segment-level reward model, which assigns a reward to each semantically complete text segment that spans over a short sequence of tokens. For reward learning, our method allows dynamic text segmentation and compatibility with standard sequence-preference datasets. For effective RL-based LM training against segment reward, we generalize the classical scalar bandit reward normalizers into location-aware normalizer functions and interpolate the segment reward for further densification. Our method performs competitively on three popular RLHF benchmarks for LM policy: AlpacaEval 2.0, Arena-Hard, and MT-Bench. Ablation studies are conducted to further demonstrate our method. Our code can be viewed at `https://github.com/yinyueqin/DenseRewardRLHF-PPO`.

---

[*]Equal contribution. This work was done during an internship at Microsoft.
[†]Corresponding Author.

# 1 Introduction

To align language models (LMs, e.g., OpenAI, 2023; Reid et al., 2024) with human values, reinforcement learning (RL, Sutton & Barto, 2018) methods have been widely adopted to optimize the non-differentiable human preference, leading to the paradigm of reinforcement learning from human feedback (RLHF, Ouyang et al., 2022; Bai et al., 2022b). A prevailing approach in RLHF is to optimize the LMs by proximal policy optimization (PPO, Schulman et al., 2017) against a *bandit* reward model learned from human preference data, with KL regularization towards a pre-specified target distribution to avoid over-optimization on the reward model (Ziegler et al., 2019; Stiennon et al., 2020; Castricato et al., 2022). While this bandit approach is easier for reward modeling and has achieved remarkable success, language generation is intrinsically sequential, rather than simultaneous. Thus, from the view of optimizing human preference, assigning a bandit reward to the entire text sequence induces the sparse reward (delayed feedback) issue (Andrychowicz et al., 2017; Marbach & Tsitsiklis, 2003), that often hurts RL-based LM training by increasing gradient variance and lowering sample efficiency (Takanobu et al., 2019; Wang et al., 2020; Guo et al., 2022; Snell et al., 2022).

To mitigate this sparse reward issue, prior works have developed methods to "ground" the sequence-level preference label into a dense token-level reward model (Yang et al., 2023; Zhong et al., 2024). While a dense per-token reward signal reduces the optimization complexity (Laidlaw et al., 2023), each action, however, is then defined as a single token, i.e., a *sub-word* that is finer-grained than a word, especially with the BPE-style tokenizers (Gage, 1994; Sennrich et al., 2016). For instance, Llama 3.1's tokenizer (Dubey et al., 2024) has tokens as {`Brit, ce, cod, neo, redd,...`} that have less clear semantic meaning *per se* in any given context. The contribution of those tokens to the text sequence will inevitably depend on later tokens, making reward/credit assignment harder, especially under the prevailing RLHF paradigm of implementing the reward model as an off-the-shelf decoder-only transformer (e.g., Ouyang et al., 2022; Bai et al., 2022b; Menick et al., 2022). Further, token-level reward implicitly assumes that the basic unit of a text sequence is *token*, which may not follow linguistics, where a more meaningful decomposition of text may be *phrase* (including *word*) that can be more semantically complete and generally consists of a short sequence of tokens.

To retain the optimization benefit of dense reward for RLHF, while mitigating its reward assignment issue and linguistic counter-intuition, in this paper, we seek to train and utilize a *segment-level* reward model, which assigns reward to each semantically meaningful text segment that constitutes a small number of (or just one) tokens. With this design, we define the action space in RLHF as "text segment," interpolating between the finest "per token" and the coarsest "full sequence" and potentially getting the benefit of both worlds: easier RL-based LM training owing to denser feedback and more accurate training guidance from the semantic completeness of each action. Although prior work has explored fine-grained RLHF through sentence-level feedback (Wu et al., 2023), such methods often rely on manual annotation or external APIs, and may miss finer compositional structures below the sentence level. In contrast, our method automatically segments text based on predictive entropy, offering fully automated, segment-level feedback from only binary sequence preferences.

Technically, we are motivated by prior works (Malinin & Gales, 2018; Li et al., 2024a) to dynamically segment a text sequence by thresholding the entropy of LM's predictive distributions, under the assumption that tokens within a semantically complete text segment can be more certainly predicted by prior tokens, while the beginning of a new segment is not (Wang et al., 2024b). To allow training the segment-level reward model by the standard sequence-preference labels via Bradley-Terry (BT, Bradley & Terry, 1952) loss, we differentiably aggregate segment rewards in a text sequence into a parametrized sequence evaluation. The learned segment-level reward model is then utilized in PPO-based policy learning, where we observe the unsuitability of classical reward normalizers, i.e., the mean and standard deviation (std) of full sequence rewards. We address this issue by generalizing the classical bandit normalizers of scalar mean and std into a mean and a std function that output the reward normalizers at arbitrary locations of the text sequence. In addition, we enhance PPO training by within-segment reward interpolation, which further densifies training signal and improves results.

We test our method on the performance of PPO-trained LM policy. On three popular RLHF benchmarks for LM policy: AlpacaEval 2.0, Arena-Hard, and MT-Bench, our method achieves competitive performance gain

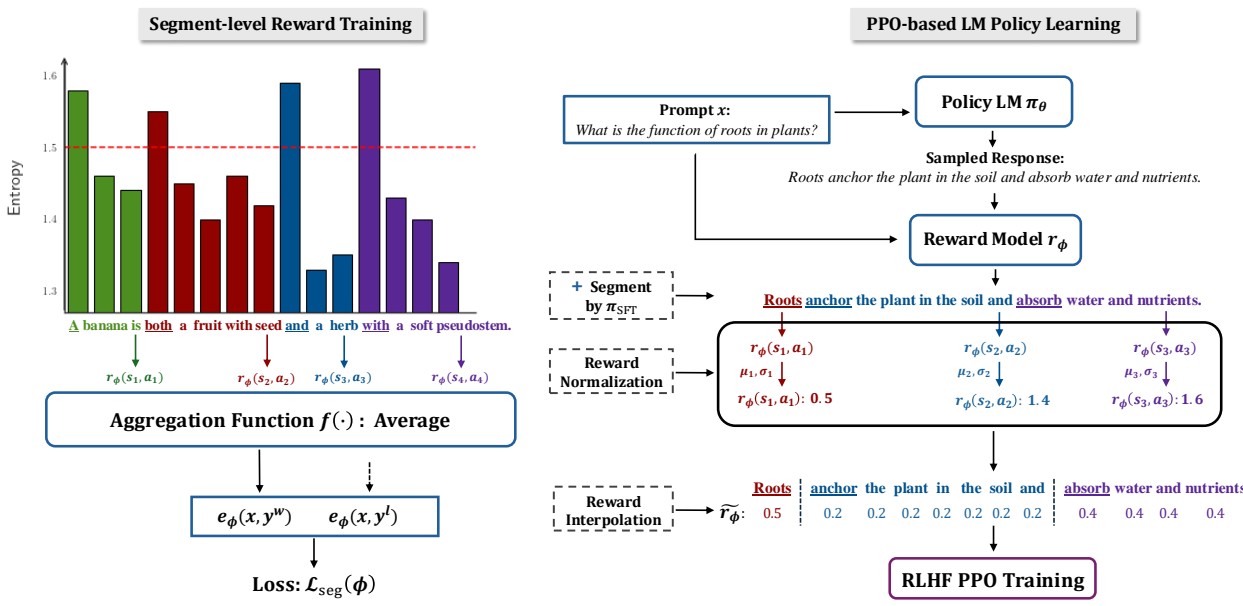

Figure 1: Overview of training and usage of our segment-level reward model. Numerical values shown are illustrative. Each text segment is represented by a different color, with its starting word underlined in the figure.

against both the classical bandit design and the recent token-level design. We conduct extensive ablation studies to verify our design choices and further probe into our method.

## 2 Main Method

### 2.1 Notations and Background

In this section, we will define generic notations, provide background on the classical bandit RLHF, and then discuss RL formulation of LM generation underlying recent efforts on dense-reward RLHF.

**Generic Notations.** Both reward modeling and LM policy learning require text prompt $x$ and the corresponding response $y$. Reward model training turns the supervised fine-tuned (SFT) model $\pi_{\mathrm{SFT}}(\cdot \,|\, \cdot)$ (without the final unembedding layer) into a parametrized scalar-output model $r_\phi(\cdot, \cdot)$ with parameter $\phi$ that scores its input. The LM policy $\pi_\theta$ is then optimized against $r_\phi$.

**Bandit Reward Model Training.** Reward model training assumes a dataset $\mathcal{D}_{\mathrm{pref}} = \{(x, y^w, y^l)\}$ of prompt $x$ and the corresponding winning/chosen response $y^w$ and losing/rejected response $y^l$, where the label comes from human evaluation on the entire text sequence $y^w$ and $y^l$. In the classical bandit RLHF, reward model $r_\phi$ is trained by the binary classification BT loss

$$\mathcal{L}_{\mathrm{bandit}}(\phi) = -\mathbb{E}_{(x, y^w, y^l) \sim \mathcal{D}_{\mathrm{pref}}} \left[ \log \sigma \left( r_\phi(x, y^w) - r_\phi(x, y^l) \right) \right], \tag{1}$$

where $\sigma(u) = 1/(1 + \exp(-u))$ denotes the sigmoid function.

**PPO-based Bandit Policy Learning.** In policy learning, a set $\mathcal{D}_{\mathrm{pol}} = \{x\}$ of text prompts $x$ is given. The LM policy $\pi_\theta$ is trained to generate outputs on $\mathcal{D}_{\mathrm{pol}}$ optimizing the bandit reward from $r_\phi$, with a KL penalty towards $\pi_{\mathrm{SFT}}$ to avoid reward over-optimization. Collectively, the objective is

$$\max_\theta \mathbb{E}_{\substack{x \sim \mathcal{D}_{\mathrm{pol}} \\ y \sim \pi_\theta(\cdot \,|\, x)}} \left[ r_\phi(x, y) - \beta \times \log \left( \frac{\pi_\theta(y \,|\, x)}{\pi_{\mathrm{SFT}}(y \,|\, x)} \right) \right], \tag{2}$$

where $\beta$ is the KL coefficient. In practice, for PPO's training stability, the value of $r_\phi(x, y)$ is de-mean and de-std normalized based on statistics calculated on a calibration dataset, e.g., $\mathcal{D}_{\text{pref}}$.

**RL Formulation of LM Generation.** By its sequential nature, LM generation is formulated as a Markov Decision Process (MDP) $\mathcal{M} = (\mathbb{S}, \mathbb{A}, P, \mathcal{R}, \gamma)$ (Sutton & Barto, 2018). Concretely, for state space $\mathbb{S}$, the state at timestep $t$, $s_t$, consists of the prompt $x$ and all generated tokens so far $a_{<t} =: [a_0, \ldots, a_{t-1}]$ with $a_{<0} =: \varnothing$, i.e., $s_t =: [x, a_{<t}]$. $\mathbb{A}$ is the action space, where the action $a_t$ at step $t$ is a short-sequence/segment of tokens from the vocabulary in our segment-level design, whereas $a_t$ is a single token in the token-level design. Transition function $P$ deterministically appends the newly sampled tokens after the previous ones, i.e., $s_{t+1} = [s_t, a_t] = [x, a_{\leq t}]$. $r(s, a) : \mathbb{S} \times \mathbb{A} \to \mathbb{R}$ scores the action choice (segment/token selection) $a$ at state/context $s$ and is typically substituted by the learned reward model $r_\phi$. $\gamma \in [0, 1]$ is the discount factor. In what follows, we will focus on our segment-level design where each action $a_t \in \mathbb{A}$ is a semantically complete text segment, consisting of a non-deterministic number of consecutive tokens. The response $y$ for prompt $x$ then contains a variable number of segments/actions, generically denoted as $y = [a_0, \ldots, a_{T-1}]$ where $T$ is the number of segments in $y$ and varies across responses. We denote a single token in $y$ as $y_i$ whose generation context is $[x, y_{<i}]$.

Fig. 1 overviews key components in our method. A detailed algorithm box is in Appendix A.

## 2.2 Reward Model Training

**Overview.** In training our segment-level reward model, we follow the data assumption set forth in Section 2.1, where the dataset $\mathcal{D}_{\text{pref}} = \{(x, y^w, y^l)\}$ contains only binary sequence-level preference labels, without any process supervision (Uesato et al., 2022). The reward model $r_\phi(s_t, a_t)$ is configured to output a scalar reward for each text segment choice $a_t$ at the generation context $s_t$. $r_\phi$ is trained such that its induced parameterized text sequence evaluations, aggregated from all segment-level rewards in the respective sequence, align with the preference labels in $\mathcal{D}_{\text{pref}}$. This is inspired by the imitation learning literature (*e.g.*, Christiano et al., 2017; Brown et al., 2019; 2020) and prior token-level reward modeling in RLHF (Yang et al., 2023). Collectively, the BT loss for training our segment-level reward function $r_\phi$ is

$$\mathcal{L}_{\text{seg}}(\phi) = -\mathbb{E}_{(x, y^w, y^l) \sim \mathcal{D}_{\text{pref}}} \Big[ \log \sigma \left( e_\phi(x, y^w) - e_\phi(x, y^l) \right) \Big],$$
$$\forall y \in \{y^w, y^l\}, \quad e_\phi(x, y) = f\big(\{r_\phi(s_t, a_t)\}_{a_t \in y}\big). \tag{3}$$

where $e_\phi$ is the parameterized sequence evaluation induced by $r_\phi$, constructed by aggregating all segment-level rewards $\{r_\phi(s_t, a_t)\}_{a_t \in y}$ in the text sequence $y$ by a selected aggregation function $f(\cdot)$.

**Entropy-based Segmentation.** As discussed in Section 1, we intend to split the given text sequence $y \in \{y^w, y^l\}$ into semantically complete segments, so that the reward assignment to each action (segment) can be easier, especially under the common implementation of the reward model as a casual LM. Recent works on LMs (*e.g.*, Li et al., 2024a; Wang et al., 2024b) have noticed that tokens within a semantically complete text segment can be more predictable by the corresponding generation context, since they are continuation of the designated semantics; whereas the starting token of a new segment is comparably less predictable, as its semantic binding with prior words is relatively weaker. For casual LMs, the predictability of each token can be conveniently measured by the entropy of the next-token-prediction distribution from which the token is sampled (Malinin & Gales, 2018). To make text sequence segmentation a one-time data pre-processing in the reward model training stage, we choose to use the prediction distribution from the supervised fine-tuned model $\pi_{\text{SFT}}$, from which the reward model is initialized before training. With a selected entropy cutoff $c_{\text{ent}}$, token $y_i$ starts a new segment if the Shannon entropy $\mathcal{H}(\cdot)$ of $\pi_{\text{SFT}}$'s predictive distribution of the $i$-th token surpasses $c_{\text{ent}}$, *i.e.*, $\mathcal{H}(\pi_{\text{SFT}}(\cdot \,|\, x, y_{<i})) > c_{\text{ent}}$, in which case $y_{i-1}$ ends the previous segment.

**Choice of the Aggregation Function $f(\cdot)$.** Aggregation function $f(\cdot)$ provides inductive bias on the relation between the quality of each segment/action and the preferability of entire text sequence. While several designs have been proposed in literature (Christiano et al., 2017; Kim et al., 2023; Yang et al., 2023), after looking into the dataset, in our experiments, we select Average to differentiably highlight the better average quality of the chosen responses over the rejected ones. With this choice of $f(\cdot)$, the parametrized sequence evaluation $e_\phi(x, y)$ in Eq. (3) is constructed as

$$e_\phi(x, y) = f(\{r_\phi(s_t, a_t)\}_{a_t \in y}) = \frac{1}{T} \sum_{t=0}^{T-1} r_\phi(s_t, a_t). \tag{4}$$

**An Alternative Interpretation.** Comparing our segment-level reward training loss Eq. (3) with the classical bandit loss Eq. (1), one may alternatively interpret $e_\phi$ and $f(\{r_\phi\})$ in Eq. (3) as a re-parametrization of the learned sequence-level feedback that differentiably aggregates the quality/contribution of each text segment, and thereby connects a denser evaluation $r_\phi$ of each semantically complete text segment with the information in ground-truth sequence-level preference label.

## 2.3 PPO-based Policy Learning

**Overview.** In policy learning, we again follow the classical bandit setting in Section 2.1 to optimize the LM policy $\pi_\theta$ on a given prompt set $\mathcal{D}_{\text{pol}} = \{x\}$. But unlike the bandit objective in Eq. (2), we adopt the full RL setting (Sutton & Barto, 2018) to maximize $\pi_\theta$'s expected sum of per-segment/step rewards. This enables directly plugging our segment-level reward model $r_\phi$ into most off-the-shelf RLHF PPO implementation. With this design, the policy learning objective for $\pi_\theta$ is

$$\max_\theta \mathbb{E}_{\substack{x \sim \mathcal{D}_{\text{pol}} \\ y \sim \prod_{t=0}^{T-1} \pi_\theta(a_t \mid s_t)}} \left[ \sum_{t=0}^{T-1} r_\phi(s_t, a_t) - \beta \log \left( \frac{\pi_\theta(y \mid x)}{\pi_{\text{SFT}}(y \mid x)} \right) \right], \tag{5}$$

where again, each $a_t$ is a segment of tokens (chopped by $\pi_{\text{SFT}}$), $s_t = [x, a_0, \ldots a_{t-1}]$ is the generation context at step $t$, and $y = [a_0, \ldots, a_{T-1}]$ is the response to prompt $x$ sampled from the learning LM policy $\pi_\theta$.

Recall from Section 2.1 that the output values from the reward model $r_\phi$ need to be normalized for the stability of PPO training. With our segment-level design, it is no longer suitable to normalize each per-step reward $r_\phi(s_t, a_t)$ by the mean and std of entire sequences' rewards as in the bandit setting, since the latter may not be on a proper scale. Further, the on-policy nature of PPO induces an extra complexity: each step of PPO samples new text sequences, whose total length, segment lengths, and segment locations are all stochastic and can differ from the reward calibration dataset, *e.g.*, $\mathcal{D}_{\text{pref}}$. Appendix H provides an extended discussion on reward normalization in PPO-based LM training. Below, we discuss our approach to construct the reward value normalizers, followed by interpolating the segment-level reward into per-token signal to helpfully provide an even denser training guidance.

**Location-aware Reward Normalizers via Regression.** While the length of the sampled response $y$ and the lengths and locations of segments $\{a_t\}$ in $y$ are all stochastic, we know that each $a_t$ is somewhere in $y$. Correspondingly, each input $(s_t, a_t)$ to $r_\phi$ is linked to a normalized location $p \in (0, 1]$ of $y$, and $p$ can be simply defined as $t/T$, where $t$ is the index of the segment $a_t$ in $y$, since PPO routine has fully sampled $y$ before calculating rewards. On each datapoint in the calibration set, normalized location $p \in (0, 1]$ again, with the linked segment-level reward available. Across all data points in the calibration set, we construct a new dataset $\mathcal{D}_{\text{norm}} = \{(p, \mu_p, \sigma_p)\}$, where $p$ runs over all values of normalized location in the calibration set, $\mu_p$ and $\sigma_p$ respectively denote sample mean and std of all segment-level rewards corresponding to $p$ in the calibration set. With $\mathcal{D}_{\text{norm}}$, we run simple linear regressions to estimate the relation between the log-transformed normalized location $\log(p)$ and the mean/std of segment-level rewards at $p$. The regression formula is given by:

$$\text{Mean}(p) = w_\mu \log(p) + b_\mu, \quad \text{Std}(p) = w_\sigma \log(p) + b_\sigma, \tag{6}$$

where the independent variable is $\log(p)$, and the regression coefficients $(w_\mu, b_\mu)$ and $(w_\sigma, b_\sigma)$ are obtained as ordinary least squares (OLS) solutions, with $\mu_p$ and $\sigma_p$ as the corresponding response variables.

Note that the classical bandit normalizers of the mean and std of full sequences' rewards correspond to evaluate $\text{Mean}(p)$ and $\text{Std}(p)$ at $p = 1.0$. In this regard, our mean and std functions in Eq. (6) generalize the classical scalar normalizers into location-aware functions able to output proper reward normalizers at an arbitrary (normalized) location $p$ of the text sequence. With $\text{Mean}(\cdot)$ and $\text{Std}(\cdot)$ and the corresponding $p$, $r_\phi(s_t, a_t)$ is normalized by $r_\phi(s_t, a_t) \leftarrow (r_\phi(s_t, a_t) - \text{Mean}(p))/\text{Std}(p)$.

**Within-segment Reward Interpolation.** Depending on the specific tokenizer in use, we observed that semantically complete text segments may contain around twenty tokens. The corresponding action space

$\mathbb{A}$ might still be large and the resulting segment-level design might not sufficiently address the sample inefficiency issue in the classical bandit RLHF and could again lead to inferior PPO-based RL training. To further densify the RL training signal, we evenly split the segment-level reward $r_\phi(s_t, a_t)$ for a segment $a_t$ to each token $y_i \in a_t$. This induces a token-level credit assignment that $\forall\, y_i \in a_t, \widetilde{r}_\phi([x, y_{<i}], y_i) = r_\phi(s_t, a_t)/|a_t|$, where $[x, y_{<i}]$ is the generation context of token $y_i$ and $|a_t|$ is the length of segment $a_t$. $\widetilde{r}_\phi$ can then directly substitute $r_\phi$ in Eq. (5), since $\sum_{t=0}^{T-1} r_\phi(s_t, a_t) = \sum_{t=0}^{T-1} (\sum_{y_i \in a_t} r_\phi(s_t, a_t)/|a_t|)$.

Note that $\widetilde{r}_\phi$ *is still intrinsically segment level*, since all token selections $y_i$ within segment $a_t$ receive the same feedback, *i.e.*, the average of segment-level reward $r_\phi(s_t, a_t)/|a_t|$. This is in contrast to prior works on token-level reward models (Yang et al., 2023; Zhong et al., 2024), where each token selection is evaluated separately and thus their token-level feedback vary for each token.

**Summary.** With the learned segment-level reward model $r_\phi$, in PPO training of the LM policy $\pi_\theta$, we first normalize each $r_\phi(s_t, a_t)$ in the sampled sequence by the corresponding normalizers $\mathrm{Mean}(p)$ and $\mathrm{Std}(p)$. Normalized segment-level rewards are then interpolated into the per-token feedback signal $\widetilde{r}_\phi$. Finally, we plug $\widetilde{r}_\phi$ directly into an off-the-shelf RLHF PPO routine. A more theoretical viewpoint of using finer-grained reward over the classical bandit reward is provided in Appendix B.

## 3   Related Work

**Training Signals for RL-based Language Model (LM) Training.** In RL-based LM fine-tuning, a classical training signal for adapting LMs to the specific downstream task is the native trajectory-level downstream test metrics (*e.g.*, Ryang & Abekawa, 2012; Ranzato et al., 2015). This approach intrinsically uses a bandit formulation of LM generation that treats the entire generated sequence as a single action. As discussed in Section 1, ignoring the sequential nature of LM generation, this bandit training signal delays the feedback to each token/phrase selection, and can thus incur optimization difficulty (Guo et al., 2022; Snell et al., 2022). With various forms of stronger data or compute requirements, task-specific per-step training signals have been proposed to mitigate this sparse reward issue. Assuming abundant golden expert data for supervised (pre-)training, Shi et al. (2018) construct per-step reward via inverse RL (Russell, 1998); Guo et al. (2018) use a hierarchical approach; Yang et al. (2018) learn LM discriminators; Lin et al. (2017) and Yu et al. (2017) use the expensive and high-variance Monte Carlo rollout to estimate per-step reward from a sequence-level adversarial reward function trained in the first place; while Le et al. (2022) use some rule-based intermediate training signal derived from the oracle sequence-level evaluation, without explicitly learning per-step reward.

Similarly, in RLHF, to move forward from the classical bandit formulation, methods have recently been proposed to ground sparse preference labels into dense per-step feedback, with applications in task-oriented dialog systems (*e.g.*, Ramachandran et al., 2021; Feng et al., 2023) and variable-length text-sequence generation (Yang et al., 2023). Recently, RTO(Zhong et al., 2024) has also explored the token-level action space for RLHF, which we adopt as one of our baselines. Our paper seeks to reconcile dense *v.s.* sparse training signal in RLHF by distributing feedback to the level of semantically complete "text segment", interpolating between the densest "token level" and the sparsest "sequence level" and ideally getting the benefit of both worlds: easier RL training and accurate optimization signal. Fine-grained rewards were also explored in Wu et al. (2023), which demonstrated their advantages over bandit rewards in detoxification and long-form QA tasks. However, their approach relies on manual segment annotation. In contrast, as shown in Section 2, our method overcomes this limitation through entropy-based automated segmentation and systematically explores the integration of segment rewards with PPO training.

In this paper, we seek to refine RL-based LM preference alignment by re-thinking the suitable action space in the RL formulation that allows both denser immediate feedback while not jeopardizing the feedback accuracy. Our segment-level design is validated through numeric and example in Section 4. We discuss a broader set of related works in Appendix G.

Table 1: Performance comparison among different action definitions on PPO-trained LM policy, with the backbone model being Phi3-mini Instruct. # {char, token} measures the average response length in the benchmark tests. Highest value of each column is in bold.

| Action Definition | AlpacaEval 2.0 | | | Arena-Hard | | MT-Bench |
|---|---|---|---|---|---|---|
| | LC(%) | WR(%) | # char | WR% | # token | GPT-4o |
| Phi3-mini Instruct | 18.89 | 14.41 | 1473 | 25.1 | 490 | 7.33 |
| Bandit (Sequence) | 27.05 | 29.07 | 2164 | 31.3 | 623 | 7.46 |
| Sentence | 25.56 | 32.92 | **2626** | 32.8 | **671** | 7.51 |
| Token | 27.82 | 26.46 | 1940 | 27.2 | 533 | 7.58 |
| Segment (**Ours**) | **31.05** | **34.53** | 2257 | **34.0** | 593 | **7.65** |
| Bandit as Segment | 14.39 | 6.46 | 691 | 11.1 | 308 | 6.61 |
| Segment as Bandit | 27.15 | 28.20 | 2079 | 30.9 | 620 | 7.38 |

## 4 Experiments

### 4.1 Experimental Setups and Implementation

**Datasets.** For reward model training, we use the preference-700K dataset[1]. which is a diverse collection of open-source preference datasets, such as HH-RLHF (Bai et al., 2022a), Stanford Human Preferences Dataset (SHP) (Ethayarajh et al., 2022), and HelpSteer (Wang et al., 2023). PPO-based LM policy training is conducted on Ultrafeedback dataset (Cui et al., 2023), from which we only use prompts to sample responses during PPO training.

**Evaluation Benchmarks.** The (PPO-trained) LM policy is evaluated on three popular open-ended instruction-following benchmarks: AlpacaEval 2.0 (Li et al., 2023), Arena-Hard (Li et al., 2024b), and MT-Bench (Zheng et al., 2023), where GPT-4o is used as the judge. For AlpacaEval 2.0, we report two metrics: Win Rate (WR), calculated as the mean of the LLM judge (GPT-4o)'s soft preference probabilities over all prompts, comparing the evaluated model to the baseline (typically GPT-4 Turbo); and Length-Controlled Win Rate (LC), which debiases for response length via logistic regression by simulating both models having equal output length, with the reported value still being the average of (adjusted) soft preference probabilities. Further evaluation details are deferred to Appendix D.

**Implementation.** We implement our method onto the open-sourced 3.8B Phi3-mini Instruct (Abdin et al., 2024), the SFT checkpoint of Phi3.1-mini Instruct, and the popular SFT checkpoint of Llama-3-8B (Dubey et al., 2024) released by RLHFlow (Dong et al., 2024)[2]. The backbone model is used as the starting points of both reward model training and PPO-based LM policy learning, in the latter initializing the models for value function, learning policy, and reference policy. Our implementation is built upon the open-source RLHF framework OpenRLHF (Hu et al., 2024). We maximally follow the default hyperparameters in OpenRLHF. Due to space limit, we defer further implementation details to Appendix D.

### 4.2 Main Experimental Comparisons

**Baselines.** To demonstrate our unique consideration of RLHF's action space, in the main experiment, we compare our design of segment-level action space with the coarsest bandit/sequence-level action space, the coarser sentence-level space, and the finest token-level space, in terms of performance of the PPO-trained LM policy. For PPO training, a reward model is first trained under the specified action definition. The sentence-level models are implemented by splitting the text sequences using sentence splitters {".", "!", "?", "\n", ";", "...", ",", ":"} and/or their foreign language equivalents. To further illustrate our segment-level reward model and denser segment-level reward assignment, we additionally compare with two hybrid approaches: *(A)* the reward model is trained using bandit-level reward (only using the EOS token),

---

[1] https://huggingface.co/datasets/hendrydong/preference_700K
[2] https://huggingface.co/RLHFlow/LLaMA3-SFT-v2

Table 2: Performance comparison among different action definitions on PPO-trained LM policies. The top four rows correspond to the 3.8B SFT checkpoint of Phi3.1-mini Instruct, and the bottom four rows correspond to the 8B SFT checkpoint of Llama-3 released by RLHFlow. Table format follows Table 1.

| Backbone Model | Action Definition | AlpacaEval 2.0 | | | Arena-Hard | | MT-Bench |
|---|---|---|---|---|---|---|---|
| | | LC (%) | WR (%) | # char | WR (%) | # token | GPT-4o |
| Phi3.1-mini-SFT | Raw Backbone | 14.93 | 10.19 | 1271 | 14.5 | 476 | 7.00 |
| | Bandit (Sequence) | 19.39 | 14.78 | 1542 | 19.5 | 524 | 7.26 |
| | Token | 22.48 | 19.25 | 1687 | 23.2 | 525 | 7.43 |
| | Segment (**Ours**) | **26.19** | **23.85** | **1795** | **28.5** | **585** | **7.49** |
| Llama-3-8B-SFT | Raw Backbone | 16.31 | 9.50 | 1221 | 10.4 | 469 | 6.82 |
| | Bandit (Sequence) | 21.20 | 20.99 | 2218 | 18.7 | 513 | 7.11 |
| | Token | 23.84 | 20.87 | 1744 | 26.0 | **622** | 7.13 |
| | Segment (**Ours**) | **25.11** | **28.57** | **2264** | **30.4** | 616 | **7.15** |

but during PPO training, rewards are computed at each segment by feeding the corresponding logits into the reward model ("Bandit as Segment"); and **(B)** the reward model is trained to produce segment-level rewards, but during PPO training, all segment rewards are aggregated into a single bandit reward applied at the EOS token ("Segment as Bandit"), where the bandit reward is implemented via the parametric sequence evaluator $e_\phi$ in Eq. (4). For all baselines, we follow the standard training receipts and tune them to the extent of ensuring a fair comparison.

**Results.** Table 1 compares our PPO-trained LM policy with alternative RLHF action spaces and two hybrid approaches using the Phi3-mini Instruct backbone. Key findings are as follows.

*(1) Our segment-level approach improves RLHF training while not suffering from length hacking.* As seen in Table 1, our LM policy performs better than the baselines across all three benchmarks: AlpacaEval 2.0, Arena-Hard, and MT-Bench. Notably, our model's average response length on AlpacaEval 2.0 and Arena-Hard is not significantly larger than the baseline models', in contrast to the LM policy from the sentence-level action space. Together, these results manifest the merit of our segment-level approach in truly improving the quality of the generated responses while not cheating the benchmark evaluations by response-length hacking (Dubois et al., 2024).

*(2) Not all finer action spaces can help RLHF training over the classical bandit formulation.* Apart from our denser segment-level approach, in Table 1, we see that the other two finer action space specifications: per-sentence and per-token, both fail to generally improve over the classical bandit/sequence-level design, especially on AlpacaEval 2.0 and Arena-Hard. This validates our design of segment-level reward assignment for RLHF PPO training, that offers more granular feedback than sentence-level and can be more accurate than the semantically incomplete token-level.

*(3) A segment-level reward model is necessary for segment-level reward assignment, and vice versa.* One may wonder if we can use the classical bandit reward model to assign segment-level reward in the PPO training. As shown by the results of "Bandit as Segment" in Table 1, this approach performs significantly worse than the original pure bandit, which in turn under-performs our segment-level design. These comparisons justify the necessity to train a segment-level reward model for segment-level reward assignment. Conversely, using our segment-level reward model to provide only bandit feedback in PPO training ("Segment as Bandit") leads to slight performance degradation over pure bandit design. Compared with our main results, we see that "Segment as Bandit" does not fully benefit from our proposal of a (consistent) segment-level action space. Its weaker results again highlight the gain of denser reward assignment in PPO-based RLHF training.

*(4) The benefit of segment-level design extends to SFT model and the larger 8B model.* We swap the backbone model to the SFT checkpoint of Phi3.1-mini Instruct and the larger 8B SFT checkpoint of Llama-3, as shown in Table Table 2. It is clear the gain of our segment-level design over the prior bandit and token-level design is not scoped within the already DPO'ed Phi3-mini Instruct. Rather, our advantage extends to both the SFT checkpoint of Phi3.1-mini Instruct and the larger Llama-3-8B-SFT, which verifies the value and versatility of our method in the practical post-training pipeline.

Table 3: Performance comparison between our segment-level PPO and DPO.

| Model | Method | AlpacaEval 2.0 (LC/WR) | MT-Bench | Arena-Hard | GSM8K |
|---|---|---|---|---|---|
| Phi3-mini Instruct | DPO | 29.24 / 29.69 | 7.27 | 28.1 | 81.6 |
| | Segment-PPO (**Ours**) | **31.05 / 34.53** | **7.65** | **34.0** | **86.7** |
| Phi3.1-mini-SFT | DPO | 24.65 / 22.73 | 7.45 | 26.6 | 82.7 |
| | Segment-PPO (**Ours**) | **26.19 / 23.85** | **7.49** | **28.5** | **85.1** |
| Llama-3-8B-SFT | DPO | 24.71 / 26.71 | 7.13 | 24.9 | 82.5 |
| | Segment-PPO (**Ours**) | **25.11 / 28.57** | **7.15** | **30.4** | **82.7** |

Table 4: Comparison of fixed $n$-gram and entropy-based segmentation on PPO-trained LM policy.

| Fixed $n$-gram | AlpacaEval 2.0 | | MT-Bench |
|---|---|---|---|
| | LC (%) | # char | GPT-4o |
| $n = 2$ | 26.00 | 2805 | 7.57 |
| $n = 5$ | 27.88 | 2224 | 7.51 |
| $n = 10$ | 28.55 | 2968 | 7.61 |
| $n = 20$ | 24.32 | **3369** | 7.58 |
| Ours | **31.05** | 2257 | **7.65** |

Appendix E provides generation examples from our main LM policy. Table 7 in Appendix C compares the LM policies in Table 1 on OpenLLM Leaderboard. Both show that our method, while achieving strong RLHF training, does not suffer from the "alignment tax" (Askell et al., 2021).

*(5) Segment-level PPO consistently outperforms DPO across models and benchmarks.* Although the primary goal of our work is to rethink the appropriate action space for PPO in RLHF, we also compare our proposed segment-level PPO with the widely adopted Direct Preference Optimization (DPO) that removes the need for an explicit reward model. As shown in Table 3, our segment-level PPO achieves stronger performance than DPO (Rafailov et al., 2023) across all metrics. In particular, it achieves up to +5.9 absolute gain on Arena-Hard (34.0 vs. 28.1) and +5.1 on GSM8K (86.7 vs. 81.6) for Phi3-mini Instruct. Similar gains are observed with the Phi3.1-mini-SFT and Llama-3-8B-SFT backbones, while DPO exhibits signs of alignment tax, particularly on GSM8K (Cobbe et al., 2021) benchmark.

### 4.3 Ablation Study

This section considers the following research questions to better understand our method. Unless otherwise specified, all ablation studies are performed on the 3.8B Phi3-mini Instruct model used in Table 1.

**(a):** *What will the performance be if we segment text by the "simpler" fixed $n$-gram?*

To compare with recent work (Chai et al., 2025), we swap our entropy-based text segmentation for the "simpler" heuristic of fixed $n$-gram, where every non-overlapping $n$ tokens in the text constitute a text segment, without considering semantics. Table 4 compares the performance of PPO-trained LM policy from our entropy-based segmentation against fixed $n$-gram with $n \in \{2, 5, 10, 20\}$.

It is clear in Table 4 that while fixed $n$-gram yields reasonable results, all of them under-performs our entropy-based segmentation, in terms of lower benchmark scores and higher response lengths. As will be discussed in the following part **(b)** and Fig. 2, our entropy-based approach segments text sequence based on semantic completeness rather than the rigid token count, which should benefit reward assignment and thus policy learning.

**(b):** *Can our method reasonably segment text and assign rewards?*

In Fig. 2 (**Top**), we compare dense reward assignments from our segment-level reward model with the token-level and fixed $n$-gram model on normal text. We choose $n$-gram with $n = 5$ as the resulted LM policy in Table 4 does not exhibit the response-length hacking issue, and so the reward model should have higher

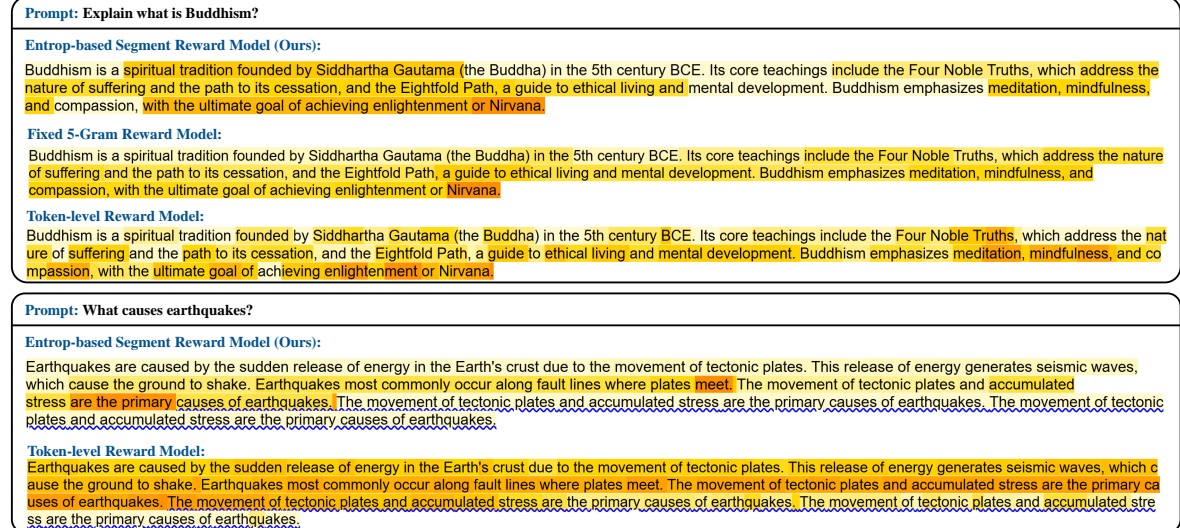

Figure 2: Examples of dense reward assignment for text sequences encountered in PPO training. In the **Top** half, we compare our segment-level reward model with the token-level and fixed $n$-gram models with $n = 5$ on normal text. In the **Bottom** half, we compare our segment-level reward model with the token-level model on text with verbosity/repetition, where repeated sentences are underlined. Darker color indicates higher reward.

Table 5: Comparison of different constructions of segment-level reward normalizers, on performance of the resulted PPO-trained LM policies.

| Reward Normalizer | AlpacaEval 2.0 | | MT-Bench |
|---|---|---|---|
| | LC (%) | # char | GPT-4o |
| No Reward Normalization | 19.64 | 2446 | 7.25 |
| Global Statistics of All | 17.34 | 2420 | 7.14 |
| Statistics of the Last Rewards | 20.30 | **2551** | 7.10 |
| Regression-based (Section 2.3) | **31.05** | 2257 | **7.65** |

quality. The color blocks in Fig. 2 (**Top**) demonstrate that our entropy-based approach segments text into meaningful semantic units. In contrast, in the token-level design, a token often represents only part of a word, and thus the reward model often inconsistently highlights only parts of words (e.g., "Tru_ths_," "medi_tation_," "comp_assion_"). The fixed $n$-gram approach rigidly segments text without considering semantics, and thus can lead to unnatural breaks, such as splitting "a guide to ethical living" into two segments: "a guide to eth" and "ical living".

In Fig. 2 (**Bottom**), we compare our segment-level reward model with the token-level model on text with verbosity/repetition. We see that our model oassigns consistent low rewards to the repeated sentences, effectively refraining the LM from verbosity. In contrast, the token-level model still assigns high rewards to tokens in the repetitions, even in the second repeat, which is undoubtedly undesirable. This comparison further shows the benefit of our design of a semantically complete action space for more accurate reward assignment.

**(c):** *How will PPO training perform if we use different constructions of reward normalizers?*

Recall that in our PPO training (Section 2.3), we use simple linear regression to fit location-aware mean and std functions that provide reward normalizers at arbitrary locations of the text sequence. To study if this design is over-engineering, we compare our main method with three simpler constructions of segment-level reward normalizers: **(A)** no reward normalization; **(B)** using the scalar global mean and std over all

Table 6: Comparison of different within-segment reward interpolation strategies. Shown are the results of the resulted PPO-trained LM policies.

| Interpolation Strategy | AlpacaEval 2.0 | | MT-Bench |
|---|---|---|---|
| | LC (%) | # char | GPT-4o |
| No Interpolation | 25.98 | **2666** | 7.45 |
| Repeat Segment Reward | 26.34 | 1795 | 7.42 |
| Even Split (Section 2.3) | **31.05** | 2257 | **7.65** |

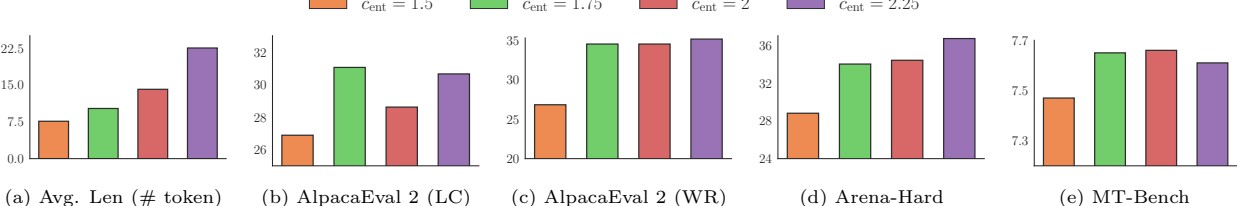

(a) Avg. Len (# token)     (b) AlpacaEval 2 (LC)     (c) AlpacaEval 2 (WR)     (d) Arena-Hard     (e) MT-Bench

Figure 3: Performance comparison among different entropy cutoffs $c_{\text{ent}}$ for entropy-based text segmentation, comparing PPO-trained LM policy's benchmark scores and average segment length ("Avg. Len") in terms of number of tokens.

segment-level rewards in the reward calibration dataset; and **(C)** using the scalar mean and std over the last segment-level rewards in each response of the calibration set, mimicking the normalizers in the classical bandit approach. Table 5 compares the resulted LM policies.

In Table 5, we clearly see that normalizing (dense) reward by improper reward statistics is akin to no reward normalization, as all three baselines have significantly lower benchmark scores than our regression-based approach and undesirable longer response lengths. As discussed in details in Appendix H, the linguistic structure of the response leads to certain correlation between the mean and std of segment-level reward values and the normalized location of segment in the response, *e.g.*, in the early or middle or later part. This necessitates our design of location-aware reward normalizers that are able to capture the reward statistics at each arbitrary location of the sampled text sequence, since constant normalization statistics can be insufficient to properly normalize the rewards of text segments at different parts of the text sequence, as verified in Table 5. Future work may extend our linear regression-based normalizer functions in Section 2.3 with non-linearity and/or more features.

**(d):** *What will happen if we use different strategies for within-segment reward interpolation?*

Recall from Section 2.3 that, to further densify the learning signal in RLHF for enhancing training, we interpolate the segment-level rewards by evenly splitting the reward of a segment to each of its constituting token. We now compare this even-split interpolation strategy with two other intuitive alternatives: **(A)** no interpolation on the segment-level rewards, use 0 for technical padding in PPO ("No Interpolation"); **(B)** repeat the segment-level reward of a segment to each token in it ("Repeat Segment Reward"). Table 6 shows the performance of the resulted PPO-trained LM policies.

In conjunction with our main result Table 1, in Table 6, we see that these two alternatives still provide (relatively) effective RLHF training on Phi3.1-mini Instruct, in reference to the results of the classical bandit approach in Table 1. Nevertheless, we see that the generation length from "No Interpolation" is significantly longer, while "Repeat Segment Reward" is too short. Probing into those long text sequences encountered in PPO training, we found that they typically contain some very negative segment-level rewards that refrains the behavior of long generation from being learned by the policy LM. Such very negative reward signals may be diluted by the technical zero-padding in "No Interpolation", leading to overly long text generation; whereas they are overly amplified in "Repeat Segment Reward", resulting in too-strong punishment for long texts and hence too-short generations. By contrast, the even-split interpolation strategy in our main method

provides densified learning signal of a proper scale, which we attribute to the implicit (segment-) length normalization inherited from the operation of dividing by segment length in an even split. Future work may design a proper non-even split of segment-level reward over each token in the text segment.

**(e):** *With a different entropy cutoff $c_{\text{ent}}$ for text segmentation, how will our method perform?*

As discussed in Section 4.1, for main results, we use entropy cutoff $c_{\text{ent}} = 1.75$ for entropy-based text segmentation. To investigate the impact of $c_{\text{ent}}$, in Fig. 3, we vary the value of $c_{\text{ent}} \in \{1.5, 1.75, 2.0, 2.25\}$, and compare the performance of the resulted PPO-trained LM policies as well as the average segment length of the PPO-trained LM policy.

As seen in Fig. 3, similar to the discussion of token-level approach in Section 1, a smaller $c_{\text{ent}} = 1.5$, which chops text sequence into finer pieces with smaller average segment length, may result in semantically less complete segments, leading to less accurate reward modeling and the subsequent weaker LM policy. A reasonably larger entropy cutoff, such as $c_{\text{ent}} \in [1.75, 2.25]$ that corresponds to an average segment length of 10 to 22 in Fig. 3a (or about 3 to 7 words), leads to much better PPO-trained LMs. This coincides with the advantage of our segment-level design over the prior token-level design in Table 1-Table 2 and verifies our goal of a more semantically complete action space.

**(f): How does our segment-based PPO policy compare on the AlpacaEval win rate-KL frontier?**

Fig. 4 shows the AlpacaEval LC win rate versus KL to the reference policy (Phi3.1-mini-SFT). For all methods, the win rate increases as KL grows, but our segment-based PPO achieves consistently higher win rates than the bandit or token-level baselines across the entire KL range. This indicates that our approach makes the most effective use of the optimization budget (KL), and sets a new frontier for real alignment performance.

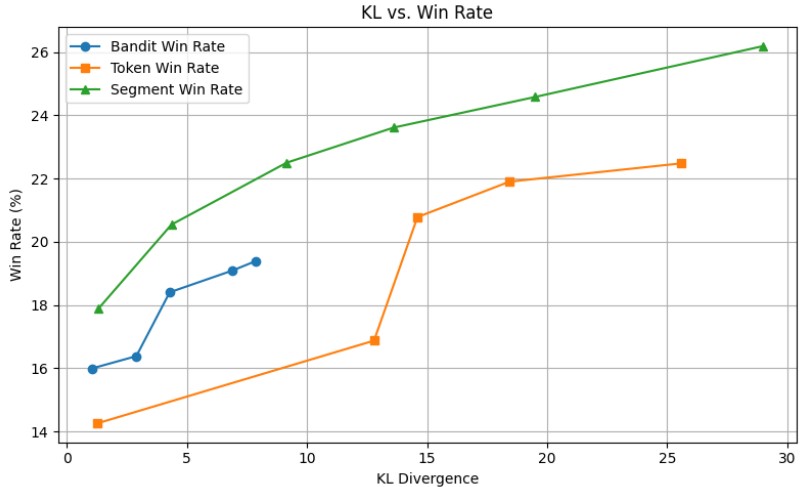

Figure 4: The AlpacaEval LC win rate-KL vs KL to the reference SFT policy.

**(g): How does segment-level reward modeling affect code generation performance?**

Figure 5 compares the performance of Bandit, Token, and Segment action spaces on three code generation benchmarks (MBPP Base (Austin et al., 2021), MBPP Plus, LiveCodeBench-V5 (Jain et al., 2024)), for both Phi3.1-mini-SFT and Llama-3-8B-SFT backbones. All models are trained on the Ultrafeedback dataset. Across all benchmarks and model backbones, our segment-based PPO consistently achieves the highest scores, surpassing both token-level and bandit baselines. This demonstrates that segment-based action space provides better and more robust performance even on structured data types such as code.

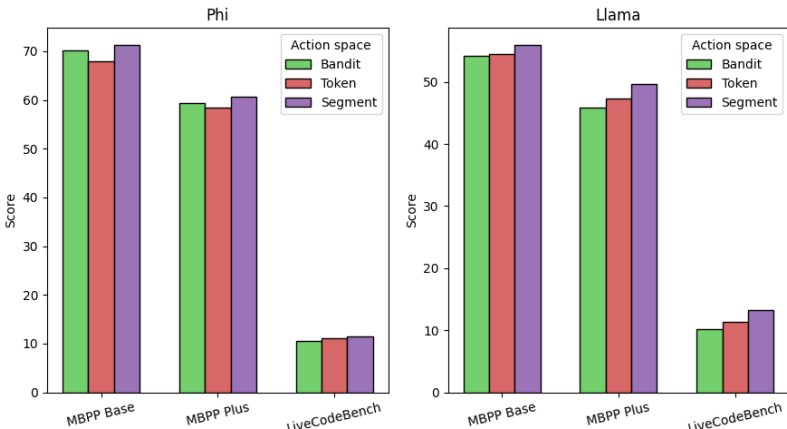

Figure 5: Performance comparison on code generation benchmarks (MBPP Base, MBPP Plus, Live-CodeBench) for different action spaces (Bandit, Token, Segment) and model backbones (Phi, Llama).

## 5 Conclusion

In this paper, we propose to train and utilize a segment-level reward model for improved RLHF in LMs, motivated by both a denser reward signal in RL-based LM training and semantic completeness of each action for accurate reward assignment. Our method and insight are validated through extensive experiments, ablation studies, and backbone models of different sizes, offering a promising research direction for further exploration of fine-grained action spaces in RLHF.

## Limitations

While our proposed segment-level reward model demonstrates promising improvements in RLHF, certain aspects warrant further investigation. As an initial exploration into refining the action space in RLHF, our experiments have so far been limited to PPO training on free-form dialog-style datasets and instruction-following benchmark evaluations. Future work will focus on scaling our approach to even larger LMs, extending its applicability to diverse tasks such as mathematical reasoning and code generation, and exploring its integration with alternative RL algorithms, such as GRPO (Shao et al., 2024), and REINFORCE++ (Hu, 2025).

## Impact Statement

Segment-PPO advances RLHF by introducing segment-level reward modeling, improving language model alignment while addressing sparse reward issues. This refinement enhances response quality, benefiting applications like conversational AI and automated content generation. However, segment-level optimization requires careful calibration to mitigate potential biases and unintended generation patterns. Additionally, as RLHF influences AI decision-making, responsible deployment is crucial to prevent misuse in misinformation propagation or biased outputs. By refining reward learning at a more semantically meaningful level, our work underscores the importance of balancing AI advancements with ethical considerations.

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

# A    Algorithm Box

Algorithm 1 summarizes our method in Section 2 on training the segment-level reward model and utilizing it in PPO-based RLHF LM training. Note that all operations in Algorithm 1 can be efficiently conducted in batch mode, parallel for multiple sample points at once.

---

**Algorithm 1** Training and Utilizing Our Segment-level Reward.

---

**Input:** Binary preference dataset $\mathcal{D}_{\text{pref}} = \{(x, y^w, y^l)\}$ for reward model training, prompt set $\mathcal{D}_{\text{pol}} = \{x\}$ for policy learning, supervised fine-tuned model $\pi_{\text{SFT}}$, reward model training steps $M_{\text{rew}}$, LM policy training steps $M_{\text{pol}}$, entropy cutoff $c_{\text{ent}}$, KL coefficient $\beta$ for RLHF PPO training.

**Initialization:** Initialize the segment-level reward model $r_\phi$ and LM policy $\pi_\theta$ from $\pi_{\text{SFT}}$, fix the aggregation function $f(\cdot)$ as the Average in Eq. (4), initialize other components in the off-the-shelf RLHF PPO routine as specified.

// Training the segment-level reward model
Use $\pi_{\text{SFT}}$ and $c_{\text{ent}}$ to split the responses $\{(y^w, y^l)\}$ in $\mathcal{D}_{\text{pref}} = \{(x, y^w, y^l)\}$ into segments.
**for** iter $\in \{1, \ldots, M_{\text{rew}}\}$ **do**
    Sample a minibatch $\mathcal{B} = \{(x_i, y_i^w, y_i^l)\}_i \sim \mathcal{D}_{\text{pref}}$.
    With $f(\cdot)$ and $\tau$, calculate $e_\phi(x_i, y_i^w)$ and $e_\phi(x_i, y_i^l)$ by Eq. (4) for $(x_i, y_i^w, y_i^l) \in \mathcal{B}$.
    Optimize reward model $r_\phi$ by Eq. (3).
**end for**

// Utilizing the segment-level reward model in PPO-based LM policy learning
Estimate the reward normalizer functions $\text{Mean}(p)$ and $\text{Std}(p)$ as described in Section 2.3.
**for** iter $\in \{1, \ldots, M_{\text{pol}}\}$ **do**
    Sample a minibatch $\mathcal{B} = \{x_i\}_i \sim \mathcal{D}_{\text{pol}}$.
    Sample a response $y_i \sim \pi_\theta(\cdot \,|\, x_i)$ for each $x_i \in \mathcal{B}$
    Use $\pi_{\text{SFT}}$ and $c_{\text{ent}}$ to segment each $y_i$; record the completion portion $p$ of each segment.
    Use $r_\phi$ to assign a segment-level reward to each segment $a_t$ in each $y_i$
    Normalize each segment reward $r_\phi(s_t, a_t)$ as $r_\phi(s_t, a_t) \leftarrow (r_\phi(s_t, a_t) - \text{Mean}(p))/\text{Std}(p)$.
    Interpolate $r_\phi(s_t, a_t)$ to each token $y_i$, as $\forall a_t \in y, \forall y_i \in a_t, \tilde{r}_\phi([x, y_{<i}], y_i) = r_\phi(s_t, a_t)/|a_t|$
    With KL coefficient $\beta$, optimize policy LM $\pi_\theta$ against $\tilde{r}_\phi$ by the PPO routine.
**end for**

---

# B    Justification of Using Finer-grained Reward over the Bandit Reward

Since our finer-grained reward is applied to the standard PPO algorithm, the standard gradient calculation of PPO should hold on our method. The benefit of fine-grained reward manifests in the estimation of value and advantage functions. For discussion/equation simplicity, below we take as an example the action-value of the first segment/action $Q(x, a_0)$ and ignore the KL-regularization term, where $x$ denotes the prompt. This discussion holds for action values at other state-actions, the value $V$ function, the advantage $A$ function, and adding back KL-regularization.

Denote $y := [a_0, a_1, \ldots, a_T]$ as the entire response, consisting of $T + 1$ segments.

$Q(x, a_0)$ **Under Bandit Reward.**    The standard bandit reward on PPO is implemented as assigning a reward $R(x, y)$ to the last segment/action, and padding all intermediate rewards as 0. With the standard selection of $\gamma = 1$ in LM's RLHF, we have

$$Q_{\text{Bandit}}(x, a_0) = \mathbb{E}_{[a_1, \ldots, a_T] \sim \pi(\cdot|s_1) \times \pi(\cdot|s_2) \times \cdots \times \pi(\cdot|s_T)} \big[ R(x, y) \big]. \tag{7}$$

$Q(x, a_0)$ **Under Segment-Level Reward.** Our denser segment-level reward model assigns a reward $r(s_t, a_t)$ to each segment $a_t$. We have

$$Q_{\text{Seg}}(x, a_0) = \mathbb{E}_{[a_1, \ldots, a_T] \sim \pi(\cdot|s_1) \times \pi(\cdot|s_2) \times \cdots \times \pi(\cdot|s_T)} \big[ r(s_1, a_1) + r(s_2, a_2) + \cdots + r(s_T, a_T) \big]. \tag{8}$$

Interchanging expectation and summation, we have

$$Q_{\text{Seg}}(x, a_0) = \mathbb{E}_{a_1 \sim \pi(\cdot \mid s_1)} \big[ r(s_1, a_1) \big] + \mathbb{E}_{[a_1, a_2] \sim \pi(\cdot \mid s_1) \times \pi(\cdot \mid s_2)} \big[ r(s_2, a_2) \big] + \cdots . \tag{9}$$

**Comparison between $Q_{\text{Bandit}}(x, a_0)$ and $Q_{\text{Seg}}(x, a_0)$.** We see that $Q_{\text{Bandit}}(x, a_0)$ is defined solely on the product space of all segments:

$$[a_1, \ldots, a_T] \sim \pi(\cdot \mid s_1) \times \pi(\cdot \mid s_2) \times \cdots \times \pi(\cdot \mid s_T). \tag{10}$$

By contrast, our $Q_{\text{Seg}}(x, a_0)$ is decomposed into $T$ terms. The first term is the expectation over a single segment, the second term over the product space of the *first two segments*, and so on. Since the sample space for estimating each term in our $Q_{\text{Seg}}(x, a_0)$ can be an order of magnitude smaller compared to $Q_{\text{Bandit}}(x, a_0)$, $Q_{\text{Seg}}(x, a_0)$ can be estimated more accurately with finite samples, which is our motivation for designing a denser reward. A formal theoretical treatment comparing dense and sparse rewards, reaching a similar conclusion, is presented in (Laidlaw et al., 2023).

Our variance reduction technique is in spirit close to the *step*-wise Importance Sampling (IS) estimator in off-policy evaluation (*v.s.* the *trajectory*-wise IS estimator). For related discussion, see Section 3.2.2 of (Jiang & Li, 2016).

## C  Additional Results

Table 7 presents the evaluation results of different LM policies from Table 1 on the HuggingFace OpenLLM Leaderboard (Beeching et al., 2023).

Table 7: Evaluation results of downstream tasks on the HuggingFace OpenLLM Leaderboard (Beeching et al., 2023), comparing LM policies in Table 1.

| Action Definition | ARC | TruthfulQA | Winograd | HellaSwag | MMLU | GSM8K | Average |
|---|---|---|---|---|---|---|---|
| Phi-Instruct | **64.76** | 54.44 | 74.51 | 79.03 | 70.41 | 81.6 | 70.79 |
| Bandit (Sequence) | **64.76** | **55.11** | 74.35 | 79.32 | 70.42 | 77.8 | 70.29 |
| Sentence | 63.40 | 53.99 | 72.93 | 79.34 | 70.42 | 84.1 | 70.70 |
| Token | 62.71 | 53.94 | 71.43 | **79.46** | **70.55** | **87.3** | 70.90 |
| Segment (**Ours**) | 62.71 | 54.74 | 72.06 | 79.23 | 70.42 | 86.7 | **70.98** |
| Bandit as Segment | 64.16 | 54.62 | 74.66 | 78.95 | **70.55** | 81.0 | 70.66 |
| Segment as Bandit | 64.33 | 54.81 | **74.74** | 79.23 | 70.39 | 78.6 | 70.35 |

## D  More Implementation Details

**Implementation Details.** We tabulate detailed parameter settings in Table 8 and Table 9. Most of them are the same as the default setting in OpenRLHF. Both the reward model and PPO training employ the Adam optimizer (Kingma & Ba, 2014), with $\beta_1 = 0.9$ and $\beta_2 = 0.95$. To save GPU memory, we use gradient checkpointing (Chen et al., 2016) and flash attention (Dao et al., 2022).

For reward model training, we set the maximum prompt sequence length as 1792 tokens, with the total sequence length (including both prompt and response) capped at 2048 tokens. During data preprocessing, we apply left truncation to the prompt and right truncation to the response. If the EOS token in the response is truncated, we manually change the last token in the truncated response to the EOS token. The

global mini batch size for reward model training is set to 128, with each GPU processing a micro batch size of 8. To facilitate distributed training, we utilize DeepSpeed ZeRO-3. For our segment-level reward model, the entropy threshold is set to $c_{\text{ent}} = 1.75$ for training with the Phi-series models and $c_{\text{ent}} = 2$ for the Llama-3-8B model. The baseline bandit reward model is technically implemented as setting the entropy threshold $c_{\text{ent}} = 1000$, restricting reward computation to the EOS token only, while the baseline token-level reward model is implemented as setting the entropy threshold $c_{\text{ent}} = 0$, ensuring that a reward is computed for each token in the text sequence.

For PPO training, the replay buffer size (`rollout_batch_size`) is set to 1024, while the batch size per GPU for generation (`micro_rollout_batch_size`) is configured as 16 for Phi-mini and 4 for Llama-3-8B. The maximum prompt sequence length is set as 1024 tokens, and the maximum generated sequence length is also set to 1024 tokens. In PPO's on-policy sampling, for each prompt in the mini-batch, a single response is sampled via top-$p$ sampling with $p = 1.0$ and sampling temperature 1.0. We use DeepSpeed ZeRO-2 for distributed training. The actor learning rate is set to the default value of $5 \times 10^{-7}$, and the critic learning rate is also the default value of $9 \times 10^{-6}$. The clipping coefficient for value loss (value clip) is set to 0.25 for PPO training based on segment- and token-level reward model, and as default to 0.2 for bandit-reward-based PPO training. The clipping coefficient for policy loss (eps clip) is set to 0.2. The KL coefficient is kept to the default value of $\beta = 0.01$.

Below, we provide further clarification of the two hybrid baselines in Table 1, which are important for interpreting our experimental results and understanding the interaction between reward model training and PPO training strategies: For the "Bandit as Segment" baseline, during reward model training, only the hidden state of the EOS token is fed into the LM head to compute the reward—hidden states at other positions are not used for supervision. However, during PPO training, instead of using the reward only at the EOS token, we feed the hidden states of all positions (not just the EOS token) into the reward model's LM head to compute per-segment rewards. This baseline illustrates that if the reward model is trained with only bandit-level supervision, but during PPO training, we attempt to provide segment-level rewards, the resulting mismatch leads to inferior performance compared to our approach. This highlights the importance of jointly considering both reward model training and PPO training strategies.

In the "Segment as Bandit" baseline, we use the same segment-level reward model as in our main PPO training. The difference is that, during PPO training, we adopt the conventional bandit action space, assigning reward only at the EOS token (i.e., only the final segment receives a non-zero reward, while all other positions receive zero). In contrast, our proposed method assigns distinct rewards to all segments. The inferior results of "Segment as Bandit" demonstrate the benefit of using segment-level rewards throughout PPO training, rather than reverting to a bandit-level reward signal.

Table 8: Hyperparameters used in reward model training.

| Hyperparameter | Value |
|---|---|
| Batch Size | 128 |
| Training Epochs | 1 |
| Max prompt length | 1728 |
| Max total length | 2048 |
| DeepSpeed ZeRO stage | 3 |
| Optimizer | Adam |
| Learning Rate | 1e-6 |
| Gradient clipping norm | 1.0 |
| Entropy threshold $c_{\text{ent}}$ | 1.75 |

# E    Examples of Generated Responses

Below, we provide examples of generated responses from our LM policy in the main results (Table 1) on the benchmarks AlpacaEval 2.0 (first two examples) and Arena-Hard (last two examples).

Table 9: Hyperparameters used in PPO-based LM training.

| Hyperparameter | Phi-mini | Llama-3-8B |
|---|---|---|
| Batch Size | 128 | 128 |
| Rollout batch size | 1024 | 1024 |
| Micro rollout batch size | 16 | 4 |
| Training Epochs | 2 | 1 |
| Max prompt length | 1024 | 1024 |
| Max generation length | 1024 | 1024 |
| DeepSpeed ZeRO stage | 2 | 2 |
| Actor learning rate | 5e-7 | 5e-7 |
| Critic learning rate | 9e-6 | 9e-6 |
| Gradient clipping norm | 1.0 | 1.0 |
| Entropy threshold $c_{\text{ent}}$ | 1.75 | 2.0 |
| Value clipping | 0.25 | 0.25 |
| KL coefficient $\beta$ | 0.01 | 0.01 |

**Prompt:** When are hops added to the brewing process?

**Response:** Hops, the flowers of the hop plant *Humulus lupulus*, are a crucial ingredient in brewing beer, primarily used for their bittering, flavoring, and aromatic properties. They are typically added to the brewing process at various stages, depending on the desired outcome in terms of bitterness, flavor, and aroma. The timing of hop addition is a critical aspect of brewing that significantly influences the final beer's character.

1. **Boiling Stage (Early Boil)**: This is the most common stage for adding hops. Adding hops early in the boil (usually within the first 15 minutes) primarily contributes to the beer's bitterness. The longer the hops are boiled, the more bitterness is extracted due to the breakdown of alpha acids, which are responsible for this bitterness. This process is known as isomerization. Early boil hops can also contribute some flavor, but the impact on aroma is minimal because the volatile compounds responsible for aroma are largely evaporated during this long boil.

2. **Middle of the Boil**: Adding hops in the middle of the boil (around 15 to 30 minutes) can strike a balance between flavor and aroma. The bitterness level is moderate, and the hop flavors and some aromatic compounds are preserved better than in the early boil, thanks to the shorter exposure time.

3. **Late Boil (Last 10-15 minutes)**: Hops added late in the boil contribute significantly to the beer's aroma and flavor, with minimal to no contribution to bitterness. This is because the shorter boiling time allows the volatile aromatic compounds to remain intact, while the alpha acids responsible for bitterness are less extracted. This technique is often used in styles that emphasize hop aroma and flavor, such as India Pale Ales (IPAs) and New England IPAs (NEIPAs).

4. **Fermentation Stage**: Some brewers also add hops during the fermentation process, a technique known as dry hopping. This is done after the primary fermentation has completed, and the beer is transferred to a secondary fermenter or directly to the bottle/keg. Dry hopping enhances the beer's aroma without significantly affecting its bitterness. The extended contact time allows the hop oils to infuse the beer, imparting a fresh, fruity, and floral aroma.

5. **Post-Fermentation (Aging)**: In some cases, especially with barrel-aged beers, hops may be added during the aging process to impart additional flavor and aroma characteristics. This can be a more experimental approach, as the interaction between the hops and the aging environment (e.g., oak barrels, different beer styles) can lead to unique and complex profiles.

The choice of when to add hops is a critical decision that brewers make based on the style of beer they are aiming to produce and the specific flavor, aroma, and bitterness profiles they desire. Experimentation

and experience play significant roles in fine-tuning this aspect of the brewing process to achieve the perfect balance for each batch.

---

**Prompt:** Introduce the "financial markets and institutions" by Frederic S. Mishkin.

**Response:** In his comprehensive work, *The Economics of Money, Banking, and Financial Markets*, Frederic S. Mishkin delves into the intricate world of financial markets and institutions, providing a thorough examination of their roles, functions, and significance in the global economy. The book, published in 2000, is a well-regarded resource for students, professionals, and policymakers alike, as it explores the complexities of financial systems and their impact on economic growth, stability, and development.

Financial markets and institutions are integral components of the broader economic system, serving as the backbone for the efficient allocation of resources, risk management, and the facilitation of transactions between various participants. Mishkin's book offers a deep dive into the various types of financial markets, such as capital markets, money markets, and foreign exchange markets, as well as the institutions that operate within these environments, including banks, investment firms, and regulatory bodies.

The book begins by establishing a solid foundation in the basic concepts and principles of money, banking, and financial markets, ensuring that readers have a clear understanding of the fundamental ideas that underpin the entire field. Mishkin then progresses to explore the role of financial intermediaries, such as banks and other institutions, in connecting savers and borrowers, and how these entities contribute to the overall efficiency and stability of the financial system.

One of the key themes in Mishkin's work is the importance of financial markets and institutions in promoting economic growth and development. He examines the ways in which these entities facilitate the flow of funds from savers to productive investments, thereby supporting innovation, entrepreneurship, and job creation. Furthermore, Mishkin discusses the role of financial markets in providing risk management tools, such as derivatives and insurance products, which help individuals and businesses protect themselves against potential losses.

Another crucial aspect of Mishkin's exploration is the examination of the regulatory framework that governs financial markets and institutions. The author emphasizes the need for effective oversight and regulation to ensure the stability of the financial system, protect consumers, and prevent financial crises. Throughout the book, Mishkin provides real-world examples and case studies to illustrate the consequences of both well-functioning and poorly regulated financial markets, highlighting the importance of maintaining a balanced and robust regulatory environment.

In addition to discussing the theoretical aspects of financial markets and institutions, Mishkin also addresses practical considerations, such as the impact of technology on the financial industry, the role of central banks in monetary policy, and the challenges posed by globalization and financial integration. By offering a comprehensive and well-rounded perspective on the subject, Mishkin's work serves as an invaluable resource for anyone seeking to understand the complexities and nuances of financial markets and institutions in the modern economic landscape.

**Prompt:** Provide me with a breakfast recipe that is quick to make and is high in protein (at least 30 grams) and has a variety of ingredients.

**Response:** Certainly! Here's a quick and delicious high-protein breakfast recipe called "Spinach and Feta Egg Muffins." This recipe is not only packed with protein but also incorporates a variety of ingredients, including spinach, feta cheese, and eggs. It's perfect for a healthy, on-the-go breakfast or a satisfying meal prep option.

**Ingredients:**

- 6 large eggs
- 1 cup fresh spinach, chopped
- 1/2 cup feta cheese, crumbled
- 1/4 cup red onion, finely chopped
- 1/4 cup cherry tomatoes, halved
- 1/4 cup green bell pepper, finely chopped
- 1/4 cup mushrooms, finely chopped
- 1/4 cup fresh parsley, chopped
- Salt and pepper, to taste
- 1/4 teaspoon garlic powder (optional)
- 1/4 teaspoon paprika (optional)
- 1/4 teaspoon dried oregano (optional)

**Instructions:**

1. Preheat your oven to 375°F (190°C) and lightly grease a muffin tin with cooking spray or a small amount of oil.

2. In a large mixing bowl, whisk the eggs until well combined.

3. Add the chopped spinach, feta cheese, red onion, cherry tomatoes, green bell pepper, mushrooms, and parsley to the egg mixture. Stir until all the ingredients are evenly distributed.

4. Season the mixture with salt, pepper, garlic powder, paprika, and dried oregano (if using) to taste.

5. Pour the egg mixture into the prepared muffin tin, filling each cup about 3/4 full.

6. Place the muffin tin in the preheated oven and bake for 20-25 minutes, or until the egg muffins are set and the tops are lightly golden.

7. Remove the muffin tin from the oven and let the egg muffins cool for a few minutes before transferring them to a wire rack to cool completely.

8. Once cooled, you can store the egg muffins in an airtight container in the refrigerator for up to 3-4 days or freeze them for up to 3 months.

9. To reheat, simply place the egg muffins in a microwave-safe dish and heat for 15-20 seconds, or until warmed through.

Enjoy your high-protein, delicious, and nutritious Spinach and Feta Egg Muffins!

**Prompt:** Write a python function that solves a quadratic equation.

**Response:** To solve a quadratic equation of the form $ax^2 + bx + c = 0$, we can use the quadratic formula:

$$x = \frac{-b \pm \sqrt{b^2 - 4ac}}{2a}$$

This formula gives us the roots of the quadratic equation, where $a$, $b$, and $c$ are coefficients of the equation, and $x$ represents the variable. The term under the square root, $b^2 - 4ac$, is known as the discriminant. The discriminant determines the nature of the roots:

- If the discriminant is positive, there are two distinct real roots. - If the discriminant is zero, there is exactly one real root (also known as a repeated or double root). - If the discriminant is negative, there are no real roots, but two complex roots.

Here's a Python function that calculates the roots of a quadratic equation, handling all three cases:

```python
import cmath

def solve_quadratic(a, b, c):
    """
    Solves a quadratic equation of the form ax^2 + bx + c = 0.

    Parameters:
    - a: Coefficient of x^2
    - b: Coefficient of x
    - c: Constant term

    Returns:
    - A tuple containing the roots of the equation. If the roots are complex,
      they are returned as complex numbers.
    """
    # Calculate the discriminant
    discriminant = cmath.sqrt(b**2 - 4*a*c)

    # Calculate the two roots using the quadratic formula
    root1 = (-b + discriminant) / (2*a)
    root2 = (-b - discriminant) / (2*a)

    return (root1, root2)

# Example usage
a, b, c = 1, 5, 6  # Coefficients for the equation x^2 + 5x + 6 = 0
roots = solve_quadratic(a, b, c)
print(f"The roots of the equation are: {roots[0]} and {roots[1]}")

# For a complex discriminant
a, b, c = 1, 2, 5  # Coefficients for the equation x^2 + 2x + 5 = 0
roots = solve_quadratic(a, b, c)
print(f"The roots of the equation are: {roots[0]} and {roots[1]}")
```

This function uses the 'cmath' module to handle complex square roots, ensuring that it can return complex roots when necessary. The function calculates and returns both roots, which can be real or complex numbers, depending on the discriminant.

**Chat Template.** Below is the chat template used for PPO training, reward model training, and evaluation sampling with Phi-3. The original Phi-3-Mini chat template includes an additional EOS token, which we removed in experiments due to incompatibility, leaving only the `<|end|>` token.

<|user|>

{the user query} <|end|>

<|assistant|>

{Assistant Response} <|end|>

For Llama-3, the chat template is as follows:

<|begin_of_text|><|start_header_id|>user <|end_header_id|>

{the user query}<|eot_id|><|start_header_id|>assistant <|end_header_id|>

{Assistant Response}<|eot_id|>

## F  Computation of Location-Aware Reward Normalizers via Regression

First, 60,000 data points are randomly sampled from the `Preference-700K` dataset, which includes pairs of prompts, chosen responses, and rejected responses. Each response is processed by a segment reward model, where the segments within the response are indexed by their respective normalized location. Specifically, the normalized location $p \in (0, 1]$ is computed for each segment $a_t$ as $p = \frac{t}{T}$, where $t$ is the index of the segment within the response and $T$ represents the total number of segments in the response. The model then provides the reward for each segment, producing a set of data points that consist of the segment's normalized location and its corresponding reward.

To estimate the relationship between the normalized location and the reward statistics, we employ a linear regression approach using the `HuberRegressor` from the `sklearn` library, which is robust to outliers. We perform the regression on the log-transformed normalized locations, $\log(p)$, to model the dependence of the mean reward $\mu_p$ and the standard deviation $\sigma_p$ of rewards at each normalized location. The regression formulas are given by:

$$\text{Mean}(p) = w_\mu \log(p) + b_\mu, \quad \text{Std}(p) = w_\sigma \log(p) + b_\sigma, \tag{11}$$

Here, $w_\mu$ and $b_\mu$ are the regression coefficients for the mean reward, and $w_\sigma$ and $b_\sigma$ are those for the standard deviation.

Once the regression coefficients are obtained, we use them to compute the normalized rewards for each segment-level reward during the PPO training. The normalized reward $r_\phi(s_t, a_t)$ is computed according to the location-aware normalizers:

$$r_\phi(s_t, a_t) \leftarrow \frac{r_\phi(s_t, a_t) - \text{Mean}(p)}{\text{Std}(p)}. \tag{12}$$

## G  More Related Work

**Reward Models in RLHF.** In the classical RLHF paradigm, policy LM is optimized against a bandit reward model trained firstly by binary classification loss on the preference dataset, with KL penalty to a specified prior distribution to avoid reward over-optimization (Ziegler et al., 2019; Stiennon et al., 2020; Jaques et al., 2020; Bai et al., 2022a; Ouyang et al., 2022). Under the same bandit formulation, recent works have enhanced the bandit reward model by directly modeling the probability of one response being preferred over the other (*e.g.*, Jiang et al., 2023; Zhao et al., 2023; Dong et al., 2024) or factorizing human preference into multiple facets via multi-objective modeling (Touvron et al., 2023; Wang et al., 2024c;a). Despite its

popularity, from the angle of RL-based optimization of human preference captured by the reward model, such a bandit reward may lead to inferior training, due to the sparse reward issue intrinsic to the bandit formulation of LM generation and credit assignment (*e.g.*, Takanobu et al., 2019; Guo et al., 2022).

Viewing the weakness of bandit RLHF, efforts have been making to densify the reward signal for RLHF LM training. Yang et al. (2023), Xu et al. (2024) and Chan et al. (2024) train token-level reward models by the binary preference classification loss. Zhong et al. (2024) and Rafailov et al. (2024) use an LM trained by DPO (Rafailov et al., 2023) firstly for token-level reward assignment, which is later used in PPO training or search-based algorithms. Guo et al. (2023), Cao et al. (2024), and Yoon et al. (2024) assign continuous or fixed fine-grained rewards (*e.g.*, $\pm 1$) by accessing an external powerful large LM or the oracle environmental reward; while Chen et al. (2024) require the extra task and datasets of erroneous solution rewriting. Apart from potential extra requirements, as discussed in Section 1, the semantic incompleteness of *token* in text may challenge the efficacy of per-token credit assignment, especially with the prevailing implementation of reward model as a decoder-only transformer that cannot look ahead into later tokens.

Close to our segment-level reward, process reward models (PRMs, *e.g.*, Uesato et al., 2022; Lightman et al., 2023) in reasoning-alike tasks also assign reward to each step, defined as a short sequence of tokens. However, *PRMs typically require per-step human annotations* – impractical for general text generation tasks like summarization or dialogue where only full text sequences can be properly evaluated. In contrast, our method (Section 2) is developed for the most basic yet general RLHF setting, where (human) preference is only provided in a dataset of binary sequence-level preference with diverse prompt-response forms.

**Learning-from-preference.** Learning-from-preference classically takes a two-stage approach where a reward model is first trained on a dataset of binary or multiple ranking via maximizing the choice model likelihood (Bradley & Terry, 1952; Plackett, 1975), before optimizing the RL/control policy against the learned reward model by RL algorithms (Akrour et al., 2011; Fürnkranz et al., 2012). Earlier application in deep learning mainly focuses on relatively simple neural-network policy for robotics/control tasks (*e.g.*, Christiano et al., 2017; Hejna & Sadigh, 2023). Implanting its success in robotics, in natural language generation, this two-stage learning-from-preference paradigm has been scaled up and popularized in the post-training stage to align LMs with specific human values, with applications ranging from text summarization (Ziegler et al., 2019; Stiennon et al., 2020), prompt generation (Yang et al., 2023), to (task-oriented) conversational agent (*e.g.*, Ouyang et al., 2022; Feng et al., 2023; OpenAI, 2023).

To alleviate the complexity in fitting an explicit reward model, motivated by the theory of maximum-entropy control and RL (Ziebart et al., 2008; Finn et al., 2016), direct preference optimization methods (DPO, *e.g.*, Rafailov et al., 2023; Zhao et al., 2023) were recently proposed to directly train LMs on a preference dataset by using their log-density-ratio as the classification logit.

## H   More on the Reward Normalizers in PPO Training

To center the assigned rewards from the reward model and reduce their variance, in most open-source (bandit) RLHF PPO implementations (*e.g.*, Havrilla et al., 2023; Hu et al., 2024), the bandit reward of the newly sampled response $y$ is first "Z-score" normalized, before being fed into the PPO routine. Concretely, for the prompt $x$ and sampled response $y$, the bandit reward $r_\phi(x, y)$ is normalized as $r_\phi(x, y) \leftarrow (r_\phi(x, y) - \mu)/\sigma$, where $\mu$ and $\sigma$ are respectively the mean and standard deviation of (bandit) rewards in the reward calibration dataset. The PPO routine starts by using this normalized $r_\phi(x, y)$, *e.g.*, first subtract it by the KL regularizer, and then calculate the advantage estimates and value function training target, *etc*.

For the segment-level action space, we will then need to normalize the reward $r_\phi(s_t, a_t)$ for each segment $a_t$. As shown in Table 5 ("Global Statistics of All"), the most intuitive idea of simply using the global mean and standard deviation over all segment-level rewards in the reward calibration dataset does not train a good LM. Looking into the responses sampled in PPO training and in the reward calibration dataset, we find that, for example, the beginning segments of the responses are typically greeting alike phrases that are less informational and/or essential to respond to the given prompt, which tend to receive relatively lower rewards. If we normalize the segment-level rewards of those early segments by the global mean and standard deviation, those normalized rewards will be significantly negative, rather than centered around 0. This will

undesirably refrain the generation of necessary greeting alike phrases, resulting in an "impolite LM" and thus inferior benchmark results. More generally, the linguistic structure of the response leads to certain correlation between the mean and standard deviation of segment-level reward values and the normalized location of segment in the response, *e.g.*, in the early or middle or later part. This observation motivates us to design location-aware reward normalizers that can approximately capture the reward statistics at an arbitrary location of the response, so that the normalized segment-level rewards can be more centered and less varying. It is important to have proper reward normalizers at an *arbitrary* location of the response, because the response sampled in PPO training will have a stochastic total length, nondeterministic number of segments, and less-controllable length of each segment. These considerations motivate our design of the regression-based reward normalizer functions in Section 2.3.

## I  More visualization results of segment

We provide additional visualization results to further demonstrate the effectiveness and interpretability of our entropy-based segmentation method. Specifically, we randomly sampled three examples from the preference-700K training set and applied our method with an entropy threshold of 1.5. As shown in Fig. 6, Fig. 7, and Fig. 8, we present direct comparisons between human-annotated segments and our automatic entropy-based segmentation, demonstrating that our approach produces semantically meaningful and consistent segments. Furthermore, Fig. 9, Fig. 10, and Fig. 11 showcase representative examples of entropy-based segmentation across diverse prompts, highlighting the robustness and adaptability of our method to various question types and response styles. Notably, the coding samples in Fig. 10 and Fig. 11 are randomly selected from the APPS training set (Hendrycks et al., 2021). In these cases, we observe that the entropy-based segmentation naturally tends to split the code at meaningful boundaries, such as line breaks (\n) or logical code blocks. This behavior indicates that to some extent, our method can automatically capture the structure and semantics of programming outputs without the need for explicit rules or supervision.

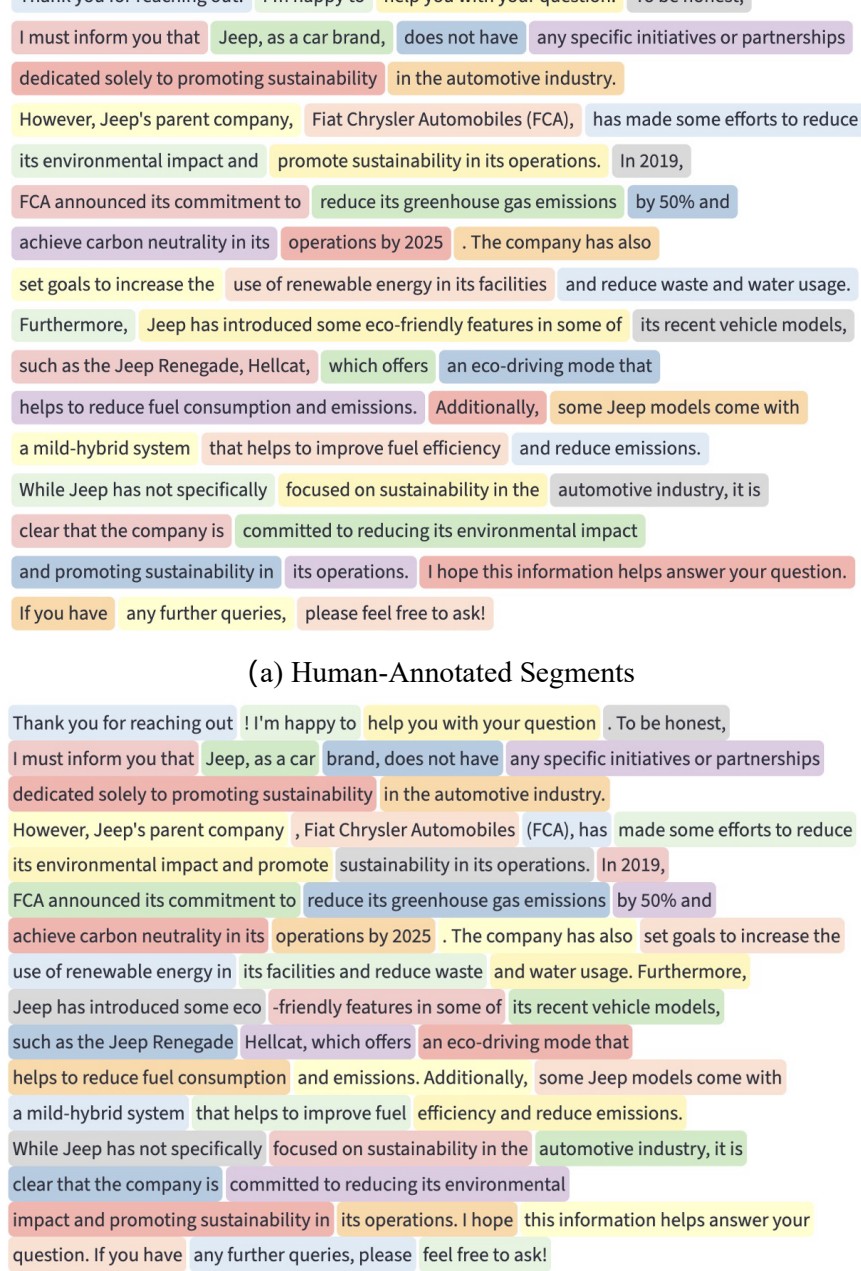

Figure 6: Visualization results comparing human-annotated segmentation (a) with entropy-based segmentation (b).

Prompt: How has the music scene in Nashville, United State changed over the past century?

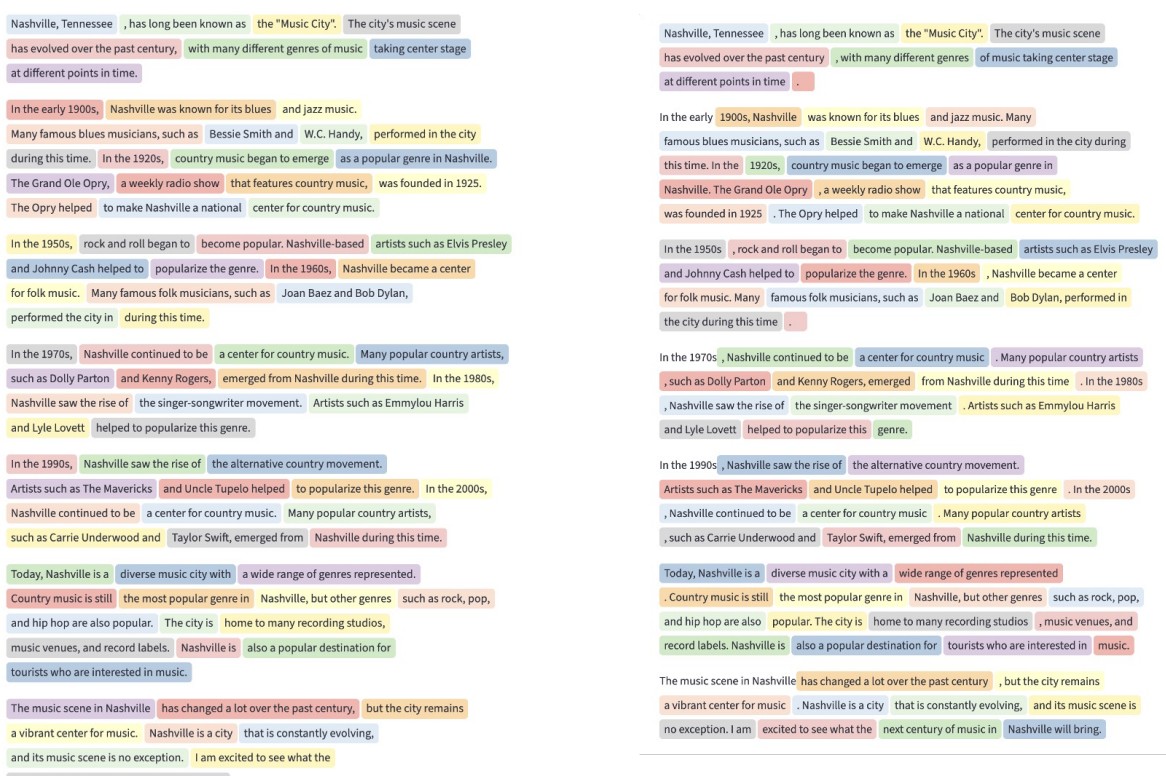

(a) Human-Annotated Segments            (b) Entropy-based Segments

Figure 7: Visualization results comparing human-annotated segmentation (a) with entropy-based segmentation (b).

Prompt: Detailed Instructions: In this task you will break down a question into the basic steps required to answer it.\n A question decomposition is a numbered list of operations that must be performed to answer the original question. Imagine explaining your question to a friendly droid by listing each action it should take in order for the question to be answered. Each step in our decomposition should refer to either an entity (known or unknown), a propery of an entity or a query operation (count, group, union, etc.)\n Here are the list of step templates and their description:\n Select: A select step is used to return a set of objects. There are no references to previous steps in a select step. template: Return [attributes]\n Filter: A filter step is used to return results from a previous step to which a certain condition applies. template: Return [#step] [condition]\n Project: A project step should return certain attributes of the results of a previous step. template: Return [attributes] of [#step]\n Aggregate: An aggregate step returns an aggregator function applied on a step's result. template: Return the [aggregator] of [#step].\n Group: A group step is an aggregator applied on attributes. template: Return the [aggregator] of [#step] for each [attribute]\n Superlative: A superlative step is used to return the result with a highest/lowest attribute among other results. template: Return [#step1] [where] [#step2] [is] [highest / lowest]\n Comparative: A comparative step is used when we need to compare an attribute with a number to filter results. template: Return [#step1] [where] [#step2] [comparator] [number] \n Union: A union step is used to return results of two steps together. template: Return [#step1] [or /,] [#step2]\n Intersection: An intersection step returns the result that two steps have in common. template: Return [attribute] of both [#step1] and [#step2]\n Discard: A discard step returns result of a step and excludes result of another step from it. template: Return [#step1] besides [#step2]\n Sort: A sort returns result of another step in a specific order. template: Return [#step1] [ordered / sorted by] [#step2]\n Is true: An is true step checks a condition on another result and returns a true or false. template: Return [is / if] [condition]\n Arithmetic: An arithmatic step operates an arithmatic operation on one or more steps. template: Return the [arithmetic op.] of [#step1] [and] [#step2].\nQ: question: How many years after the Treaty of Verdun did Philip IV of France become King?

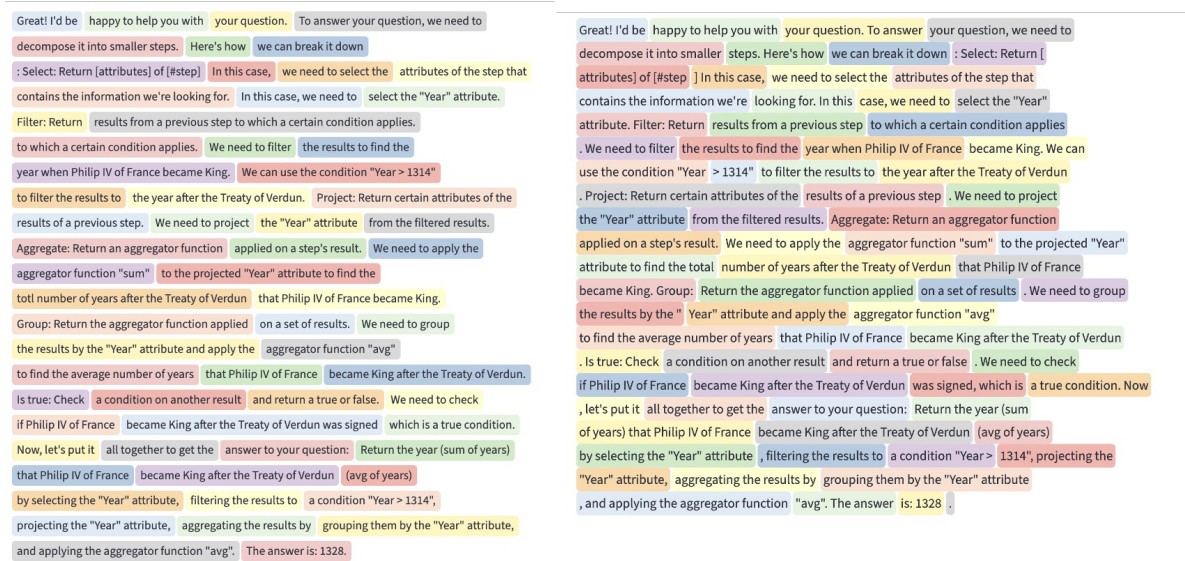

(a) Human-Annotated Segments

(b) Entropy-based Segments

Figure 8: Visualization results comparing human-annotated segmentation (a) with entropy-based segmentation (b).

Part #1: Translate a text from English to Urdu . Usage: "The Survey further notes that India 's energy intensity of GDP started declining at a much lower level of per capita GDP as compared to the developed world بالکل مختصر طور ونے لگی شدت کم ہ یشت کی توانائی بھارت کی مع ہ دیتا ہے کہ بھی حوال مزید یہ سروے" ﺣ". Explanation".ممالک ترقی یافتہ میں دنیا کے مقارنے معیشت کے فی شخص مضبوط پر کی طرف سے : The original text is translated from English to Urdu , so it meets the criteria.

Part #2: The translation must not omit or add information to the original sentence حوالہ دیتا ہے کہ بھارت کی معیشت کی توانائی شدت کم ہونے لگی سروے مزید یہ بھی" text . Usage: The translated یافتہ ممالک مضبوط معیشت کے مقارنے میں دنیا کے ترقی طرف سے فی شخص بالکل مختصر طور پر کی ." contains all the key information from the original sentence without any omissions or additions. The translation specifically mentions the decline of India's energy intensity of GDP, the comparison to the developed world , and that it started at a lower per capita GDP. Explanation: The translated text contains all the information from the original sentence , so it meets the criteria.

Part #3 : The translation must convey the same meaning as the original sentence. Usage: سروے مزید یہ بھی حوالہ دیتا ہے کہ بھارت کی معیشت کی توانائی شدت کم ہونے لگی بالکل" The translated text یافتہ ممالک بوط معیشت کے مقارنے میں دنیا کے ترقی مختصر طور پر کی طرف سے فی شخص مض". conveys the same meaning as "The Survey further notes that India's energy intensity of GDP started declining at a much lower level of per capita GDP as compared to the developed world ." Explanation: The translated text conveys the same idea and meaning as the original sentence, so it meets the criteria.

Figure 9: Example of entropy-based segmentation.

```
["import sys\ninput = sys.stdin.readline\n  \n\nt = int(input())\nfor
 _ in range(t):\n n = int(input())\n a = list(map(int, input().split()))\n  \n ans = 0\n
for i in range(n - 1):\n diff = a[i] - a[i + 1]\n if diff ≤ 0:\n continue\n  else:\n ans = max(len
(bin(diff)) - 2, ans)\n  a[i + 1] = a[i]\n print(ans)\n
", "for t in range(int(input())):\n n = int(input())\n a = [int(i) for i in input().split()]\n  m = a[0]\n  v = 0\n
for i in a:\n v = max(v,m  -i)\n m = max(m,i)\n if v  == 0:\n print(0)\n else:\n  p = 1\n  c = 0\n  while p<=v:\n
p *= 2\n c += 1\n  print(c)\n", "
for _ in range(int(input())):\n n = int(input())\n l1 = list(map(int, input().split()))\n  answer = 0\n
max_so_far = l1[0]\n  for i in range(1, n):\n  if l1[i] ≥ max_so_far:\n max_so_far = l1[i]\n else:\n
answer = max(answer,  (max_so_far - l1[i]).bit_length())\n  \n \n print(answer)"]
```

Figure 10: Example of entropy-based segmentation on coding response.

["n = int(input()) \na, = map(int, input().split())\ndp = [[ 0 for i in range(n + 1)] for j in range(n + 1)]\n for i in range(n):\n dp[0][i] = a[i]\nfor i in range(1, n):\n for j in range(n - i + 1):\n dp[i][j] = dp[i - 1][j] ^ dp[i - 1][j + 1]\n for i in range(1, n):\n for j in range(n - i):\n dp[i][j] = max(dp[i][j], dp[i - 1][j], dp[i - 1][j + 1])\nfor i in range(int(input())):\n l, r = map(int, input().split())\n print(dp[r - l][l - 1]) ", "# import os, sys, atexit\n # from io import BytesIO, StringIO\nfrom sys import stdin, stdout\n \n# input = BytesIO(os.read(0, os.fstat(0).st_size)).readline\n # _OUTPUT_BUFFER = StringIO()\n# sys.stdout = _OUTPUT_BUFFER\n \n# @atexit.register\n# def write():\n# sys.stdout.write(_OUTPUT_BUFFER.getvalue())\n \ndef calculate(array):\n n = len(array)\n finalarray = []\n finalarray.append(array)\n finalarray.append([])\n while (n!=1):\n for x in range(n-1):\n finalarray[-1].append(finalarray[-2][x]^finalarray[-2][x+1])\n finalarray.append([])\n n -=1\n return finalarray\n \ndef solve():\n n = int(input())\n array = [0]\n array.extend(list(map(int,stdin.readline().strip().split())))\n subArrays = []\n for x in range(n+1):\n subArrays.append([0 ] (n+1))\n finalarray = calculate(array)\n # print (finalarray,len(finalarray))\n for x in range(1,n+1):\n for y in range(x ,n+1):\n # print (y-x+1 ,x,finalarray[y-x+1])\n value = finalarray[y-x][x]\n subArrays[1][y] = max(subArrays[1][y],value)\n subArrays[x][y] = max(subArrays[x][y],value)\n # print (subArrays)\n for x in range(1,n+1):\n for y in range(2,n+1):\n subArrays[x][y] = max(subArrays[x][y],subArrays[x][y-1])\n # print (subArrays)\n for y in range(1,n+1):\n for x in range(n -1,0,-1):\n subArrays[x][y] = max(subArrays[x][y],subArrays[x+1][y])\n # print (subArrays)\n q = int(input())\n for _ in range(q):\n l, r = list(map(int,stdin.readline().strip().split()))\n print(subArrays[l][r]) \ntry:\n solve()\n except Exception as e:\n print (e)\n\n # solve()\t\t \t\t\t \t \t \t\t\t\t \t\t\t\n ", "#t=int(input())\nt =1\nfor _ in range(t):\n n=int(input())\n l=list(map(int,input().split()))\n dp=[[0 for j in range(n)] for i in range(n)]\n for i in range(n):\n dp[0][i]=l[i]\n for i in range(1,n):\n for j in range(n-i):\n dp[i][j]=dp[i-1][j]^dp[i-1][j+1]\n \n for i in range(1,n):\n for j in range(n-i):\n dp[i][j]=max(dp[i][j],dp[i-1][j],dp[i-1][j+1])\n \n q=int(input())\n for __ in range(q):\n x,y=map(int,input().split())\n x-=1\n y-=1\n print(dp[y-x][x]) ", "import sys\n input=sys.stdin.readline\n n=int(input())\nar=list(map(int,input().split())) ndic={}\n li=[]\nfor i in range(n):\n xx=[]\n for j in range(n-i):\n xx.append(0)\n li.append(xx.copy())\n for i in range(n):\n for j in range(n-i):\n if (i==0):\n li[i][j]=ar[j]\n else:\n li[i][j]=li[i-1][j]^li[i-1][j+1]\n for i in range(1,n):\n for j in range(n-i):\n li[i][j]=max(li[i][j],li[i-1][j],li[i-1][j+1])\nfor _ in range(int(input())):\n l,r=map(int,input().split())\n print(li[r-l][l-1]) ", "#!/usr/bin/env python3\n #\n# XOR-pyramid\n#\nimport sys, os\n\ndef read_int(): return int(input())\ndef read_ints(): return list(map(int, input().split()))\n#-----------------------------------------------------------------------------#\n\n\n = read_int() na = read_ints()\ndp = [[0] * n for _ in range(n)]\nf = [[0] * n for _ in range(n)]\n for i in range(n - 1, -1, -1):\n\t f[i][i] = a[i]\n\tfor j in range(i + 1, n):\n\t\t f[i][j] = f[i][j - 1] ^ f[i + 1][j]\n for i in range(n - 1, -1, -1):\n\tdp[i][i] = f[i][i]\n\tfor j in range(i + 1, n):\n\t\tdp[i][j] = max(f[i][j], dp[i][j - 1], dp[i + 1][j])\nq = read_int()\nfor _ in range(q):\n\tl, r = read_ints()\n\tprint(dp[l - 1][

Figure 11: Example of entropy-based segmentation on coding response.

