# OpenReview forum: "Segmenting Text and Learning Their Rewards for Improved RLHF in Language Model"
_TMLR — Accepted by TMLR_

### Review · Reviewer_3Cy9 · 2025-07-15

**Summary Of Contributions:**

The authors address the challenge of reward sparsity in RLHF training by proposing a segment-level reward assignment. They argue that in standard RLHF approaches, which are typically based on a bandit formulation, rewards are only assigned after the end-of-sequence (EOS) token, resulting in a sparse feedback. While token-level rewards could provide more dense signals, current tokenization methods often generate tokens that are not semantically meaningful, making this approach suboptimal. To mitigate this, the authors introduce segment-level credit assignment, where segments are defined by thresholding on entropy. Empirical evaluations on AlpacaEval 2.0, Arena-Hard, and MT-Bench demonstrate the effectiveness of segment-level rewards over both sequence-level and token-level credit assignment strategies.

**Audience:**

Yes

**Claims And Evidence:**

Yes

**Requested Changes:**

1.  It is important to visualize the relationship between performance and KL divergence. Specifically, the authors should focus on analyzing reward vs. KL or win-Rate vs. KL trade-off for the different baseline methods. This would provide clear evidence of efficiency of the proposed segment-level training.

2. The current evaluation focuses primarily on instruction-following benchmarks like AlpacaEval and MT-Bench. Evaluation on coding instruction prompts or specifically multi-step reasoning tasks would be interesting.

3. I understand due to space constraints the evaluations metrics have been deferred to Appendix. For improved readability, the result tables in the main paper should be more self-contained. Currently, metrics such as "LC" and "WR" are used in Tables 1, 2, and 3 , but their definitions are deferred to Appendix D.

**Strengths And Weaknesses:**

>**Strengths:**

1. The paper is generally well-written and easy to follow. The research problem and motivation has been clearly articulated in the introductions.
2. The core idea of using segment-level reward assignment is interesting. Further, each component of the proposed approach such as segmentation technique, reward normalization etc has been clearly explained.

3. The empirical evaluations seems comprehensive in terms of models and datasets. The authors performed several ablations, which provide useful insights.


>**Weaknesses:**

1. The proposed method is evaluated exclusively within a PPO-based RLHF framework. While the paper shows better performance compared to a baseline DPO model, it is unclear if the segment-level reward modeling concept could be adapted to offline algorithms such as DPO.

2. The text segmentation is based on the entropy of the SFT model, both for reward model training and during PPO training. The rationale for not using the current learning policy's entropy for segmentation is not fully justified. Since responses are sampled from the current policy during PPO, using its entropy for segmentation seems more intuitive.

3. The paper does not have any analysis on the KL divergence between the trained policy and the SFT model.

4. Based on current setup, it is difficult to comprehend how the segment-level rewards are distributed for codinginstruction prompts. The authors should also add evaluation for these kind of prompts.

>**Questions:**

1. "after looking into the dataset, in our experiments, we select Average to differentiably highlight the better average quality of the chosen responses over the rejected one." -- Does this imply that the choice of the aggregation function is a dataset-dependent hyperparameter that may require manual tuning for different datasets?

2. The win-rate in Tables 1 and 2 appear somewhat low. Is the win-rate relative to SFT policy? How is the win-rate calculated: (#win + #tie/2)?

---

> ### Author Response · Authors · 2025-07-24
> **Response to Reviewer 3Cy9**
>
> We thank Reviewer **3Cy9** for constructive feedback. Below are our detailed responses to your concerns.
>
>
>
> > **W1.**
> Whether the segment-level reward modeling can be adapted to DPO.
>
> We thank the reviewer for this valuable suggestion. In this work, our primary focus is to investigate how different definitions of the action space and reward credit assignment in PPO training—specifically, the segment-based action space, together with our proposed location-aware reward normalizers via regression and within-segment reward interpolation—jointly contribute to the improved performance.  Adapting part of our approach (the segment-based action space) to DPO is indeed an interesting direction. However, a detailed exploration of how segment-level reward modeling could be effectively integrated into DPO falls outside the scope of the current paper.
>
>
> > **W2.**
> Clarification for using SFT model for text segmentation.
>
> We thank the reviewer for this insightful suggestion. We use the SFT model’s entropy for segmentation to ensure stability and consistency throughout both reward model training and PPO. If segmentation were based on the current policy’s entropy, the segment boundaries could shift as the policy evolves, introducing instability in reward assignment and credit allocation.
>
>
> > **W3 && R1.**
> Analysis on the KL divergence between the trained policy and the SFT model.
>
> Thank you for the valuable suggestion. We have added an analysis of the KL divergence between the trained policy and the SFT model in the ablation section of the manuscript (see Figure 4). The newly included AlpacaEval win rate vs. KL curves show that our segment-based PPO policy consistently achieves higher win rates than both the bandit and token-level baselines at the same level of KL divergence. This demonstrates that our approach makes more effective use of the optimization budget and sets a new frontier in alignment performance.
>
> > **W4 && R2.**
> The evaluation on coding tasks.
>
> Thank you for the valuable suggestion. We have added evaluation results on coding benchmarks (see Figure 5), which show that segment-based PPO also achieves better performance on these structured tasks. In addition, Figure 10 and Figure 11 in the appendix present visualizations of ground-truth responses sampled from the APPS coding dataset and their corresponding entropy-based segmentations. As illustrated, our entropy-based segmentation method naturally tends to split code at meaningful boundaries, such as line breaks (\n) or within complete code blocks.  Exploring additional training, evaluation, and analysis on more types of structured data is an interesting direction, which we leave for future work.
>
>
> > **Q1.**
> The choice of aggregation function.
>
> Thank you for the question. We have experimented with both mean (average) and sum aggregation functions and observed comparable performance between the two in our experiments. While the choice of aggregation function could, in principle, be treated as a dataset-dependent hyperparameter, our results suggest that the method is robust to this choice. For simplicity and ease of interpretation, we report results with the mean aggregation.
>
>
>
> > **R2.**
> Definition of metrics such as "LC" and "WR".
>
>
> Thank you for your comment. For better readability, we have now moved the definitions of Win Rate (WR) and Length-Controlled Win Rate (LC) to Section 4.1 of the main text:
>
> Win Rate (WR): Calculated as the mean of the LLM judge (GPT-4o)’s soft preference probabilities over all prompts, comparing the evaluated model to the baseline (typically GPT-4 Turbo, not an SFT policy).
>
> Length-Controlled Win Rate (LC): Debiased for response length using logistic regression, simulating both models having equal output length. The reported value is still the average of (adjusted) soft preference probabilities.

---

> > ### Comment · Reviewer_3Cy9 · 2025-08-12
> > **Thank you for your response**
> >
> > Thank you for the detailed discussion and for clarifying all concerns. With the updated changes: additional KL plots and evaluation results on coding benchmarks, the paper justifies all claims.

---

### Review · Reviewer_VeAx · 2025-07-16

**Summary Of Contributions:**

This paper presents a novel approach to improve RLHF by addressing the sparse reward problem inherent in traditional bandit-based formulations. Instead of assigning a single reward to an entire generated sequence, the authors propose a segment-level reward model. This model assigns rewards to semantically coherent text segments, which are sequences of tokens that are more meaningful than individual tokens but more granular than the full text. The paper introduces a method to dynamically segment text by thresholding the entropy of the language model's predictive distribution, based on the idea that beginnings of new segments are less predictable. To train this segment-level reward model using standard sequence-preference datasets, the authors aggregate the individual segment rewards into a single sequence-level score. For the policy learning phase using PPO, the paper introduces location-aware reward normalizers, which are functions that adjust the reward normalization based on the segment's position within the sequence. This is achieved by learning a regression from the normalized location of a segment to the mean and standard deviation of rewards. Furthermore, the segment-level reward is interpolated across its constituent tokens to create an even denser training signal. The authors validate their method on several RLHF benchmarks, including AlpacaEval 2.0, Arena-Hard, and MT-Bench, and show that their approach outperforms both traditional sequence-level and more recent token-level RLHF methods.

**Audience:**

Yes

**Broader Impact Concerns:**

None.

**Claims And Evidence:**

Yes

**Requested Changes:**

- The core assumption of the paper is that text can be segmented into "semantically complete" units by thresholding the entropy of the language model's predictive distribution. This is stated in Section 2.2: "Recent works... have noticed that tokens within a semantically complete text segment can be more predictable... whereas the starting token of a new segment is comparably less predictable". While this is an intuitive and elegant heuristic, it remains an assumption. To make the paper's primary contribution more robust, I suggest the authors to provide a more rigorous justification for this link. Proposed Action: I recommend conducting a small-scale qualitative analysis. The authors could randomly sample 50-100 responses from a test set, have a human annotator segment them based on semantic completeness, and then compare these manual segmentations to the ones generated by the entropy-based method. Providing a few illustrative examples in the Appendix, showing where the method succeeds and perhaps where it fails, would substantially strengthen the foundation of the work. This would provide direct evidence supporting the central hypothesis beyond the (strong) downstream results.


- The two hybrid baselines in Table 1 are important for the paper's argument, but their descriptions are very brief. For instance, for "Bandit as Segment," it is stated that "...rewards are computed at each segment by feeding the corresponding logits into the reward model". It is not perfectly clear what "logits" are being referred to here (e.g., the logits of the tokens in the segment? An aggregated representation?). Proposed Action: I suggest adding a few more precise details in Section 4.2 or the Appendix explaining the exact inputs to the reward model for these two specific baselines. This would improve the reproducibility and clarity of these important comparisons.

**Strengths And Weaknesses:**

Strengths :-

- The core idea of segmenting text into semantically meaningful chunks offers a compelling middle ground between the coarse sequence-level and the often oversubtle token-level reward assignments.
- The proposed method demonstrates superior performance across multiple benchmarks (AlpacaEval 2.0, Arena-Hard, and MT-Bench) when compared to established baselines like bandit, sentence-level, and token-level RLHF.
- The use of predictive entropy for text segmentation is an elegant and fully automated solution, which circumvents the need for manual annotation or external tools that other fine-grained RLHF methods might require.
- The paper shows that their segment-level PPO approach consistently outperforms DPO, a widely adopted alternative to RLHF that does away with an explicit reward model. For example, on Phi3-mini Instruct, Segment-PPO achieves an 86.7 on GSM8K, compared to DPO's 81.6 (Table 3). This provides strong evidence for the continued relevance of PPO-based methods when coupled with a well-designed reward structure.

Weaknesses :-

- The paper's method for segmentation hinges on the assumption that lower entropy in a language model's predictive distribution corresponds to a "semantically complete text segment." While this is a reasonable heuristic, the paper does not provide a rigorous justification for this link. It is possible that low entropy could also be a result of other factors, such as repetitive or formulaic language, and may not always align with true semantic boundaries. The paper states, "...tokens within a semantically complete text segment can be more predictable by the corresponding generation context..." (Section 2.2), but this remains an assumption.

- The authors claim their method improves quality without "cheating the benchmark evaluations by response-length hacking" (Section 4.2). However, the mechanism for how segment-level rewards might influence the length of generated text is not fully explored. Since the final reward is an average of segment rewards, this could inadvertently incentivize the model to produce a larger number of shorter, high-reward segments, which might not always lead to a better overall response. A more detailed analysis of the relationship between segment length, reward, and output length would strengthen this claim.

-  The paper proposes using a simple linear regression on the log-transformed normalized location to create location-aware reward normalizers. The paper acknowledges this limitation, stating, "Future work may extend our linear regression-based normalizer functions in Section 2.3 with non-linearity and/or more features" (Section 4.3, part c). While the linear model shows good results, it may not capture more complex relationships between a segment's position and its expected reward, especially in long or structurally complex responses.

- The decision to evenly split the segment-level reward among its constituent tokens is another heuristic choice. The paper notes, "Future work may design a proper non-even split of segment-level reward over each token in the text segment" (Section 4.3, part d). This even split ignores the possibility that some tokens within a segment might be more critical to its overall quality than others.

- While the benchmarks used are standard for instruction-following and dialogue tasks, the paper's claims would be more robust if the method were tested on a wider range of applications. The limitations section acknowledges this, stating, "Future work will focus on scaling our approach to even larger LMs, extending its applicability to diverse tasks such as mathematical reasoning and code generation..." (Section 5). The current evaluation does not provide evidence that the entropy-based segmentation and reward model would be effective in these more structured domains.

---

> ### Author Response · Authors · 2025-07-24
> **Response to Reviewer VeAx**
>
> We would like to thank Reviewer **VeAx** for the careful review.
> The following are our responses to your questions.
>
>
> > **W1 && R1.**
> Justification of Entropy-based Segmentation.
>
> We appreciate your concern regarding the potential issue that, when evaluating generated responses, repeated sentences could lead to artificially low entropy. However, in our work, entropy-based segmentation is applied during the training process—specifically, when training the reward model—using the chosen and rejected responses from the preference700k dataset. We randomly sampled and manually checked examples from this dataset and found that repeated sentences are very rare in these responses. Therefore, using entropy for segmentation is reasonable in our training setting.
>
> In response to your suggestion, we conducted a qualitative analysis as shown in Appendix I. Figures 6, 7, and 8 compare human and entropy-based segmentations on sampled responses from preference700k. Most cases show strong alignment, e.g., at phrase or clause boundaries. Occasionally, as in Figure 8b, the method splits a unit like “Year > 1314” into “Year” and “> 1314”—where ideally they would remain together as a single segment. Nevertheless, the majority of segmentations are reasonable. Exploring even better automatic segmentation approaches is an interesting direction for future work.
>
> Thank you again for your constructive suggestion, which helped us strengthen our empirical validation.
>
>
> > **W2.**
> The response length.
>
> We apologize for any potential misunderstanding. The referred sentence was written under the specific context of Sec 4.2. Concretely, in our main result (Table 1), we observe that in both AlpacaEval 2.0 and  Arena-Hard, our model’s response length with the baselines Sequence/Sentence/Token-level reward, while having higher benchmark scores, which verify that the gain of our method over the baselines truly comes from quality improvement, rather than reward hacking. In our main experiment, the average segment length is 8.5 tokens, which is in the middle ground.
>
> Table 1 also infer that there is little incentivization for the model to produce many short segments, as the token-level reward variant does not have very long response length. Table 4 (Sec. 4.3 (a)) presents the AlpacaEval 2.0 and MT-Bench scores for the Fixed n-gram segmentation method (with $n \in \{2, 5, 10, 20\}$ and our main results (avg segment length: 8.5 tokens). We see from Table 4 that the fixed n-gram with n = 10 is the best  n-gram variant. Additionally, Fig 3 (Sec. 4.3 (e)) compares the performance of our method under various entropy cutoff (equiv. to various avg segment length), again showing that a middle-ground avg segment length of 10 tokens is the most performant.
>
> > **W3 && W4.**
> Two Practical Strategies for PPO Stability.
>
> We acknowledge that both our location-aware reward normalizer (using linear regression) and the even-split within-segment reward interpolation are relatively straightforward choices. However, our ablation studies in Section 4.3 (Tables 5 & 6) demonstrate that these simple strategies are not only effective but also outperform alternative methods such as global normalization, normalization using only last-segment statistics, or other interpolation schemes.
>
> Our results suggest that these approaches provide the best balance of stability and performance for PPO training in our experiments. We agree that exploring more advanced (e.g., nonlinear or feature-rich) normalization or reward assignment strategies could further improve results, and we have highlighted these as promising directions for future work.
>
>
>
> > **W5.**
> Evaluation on Structured Domain Tasks.
>
> Thank you for the valuable suggestion. We have added evaluation results on coding benchmarks (see Figure 5), which show that segment-based PPO also achieves better performance on these structured tasks. In addition, Figure 10 and Figure 11 in the appendix present visualizations of ground-truth responses sampled from the APPS coding dataset and their corresponding entropy-based segmentations. As illustrated, our entropy-based segmentation method naturally tends to split code at meaningful boundaries, such as line breaks (\n) or within complete code blocks.  Exploring additional training, evaluation, and analysis on more types of structured data is an interesting direction, which we leave for future work.

---

> > ### Author Response · Authors · 2025-07-24
> > **Response to Reviewer VeAx**
> >
> > > **R2.**
> > Justification of Two Baselines Description.
> >
> > Thank you for highlighting the need for more precise descriptions of the “Bandit as Segment” and “Segment as Bandit” baselines.
> > For the “Bandit as Segment” baseline, during reward model training, only the hidden state of the EOS token is fed into the LM head to compute the reward—hidden states at other positions are not used for supervision. However, during PPO training, instead of using the reward only at the EOS token, we feed the hidden states of all positions (not just the EOS token) into the reward model’s LM head to compute per-segment rewards. This baseline illustrates that if the reward model is trained with only bandit-level supervision, but during PPO training, we attempt to provide segment-level rewards, the resulting mismatch leads to inferior performance compared to our approach. This highlights the importance of jointly considering both reward model training and PPO training strategies.
> >
> > In the “Segment as Bandit” baseline, we use the same segment-level reward model as in our main PPO training. The difference is that, during PPO training, we adopt the conventional bandit action space, assigning reward only at the EOS token (i.e., only the final segment receives a non-zero reward, while all other positions receive zero). In contrast, our proposed method assigns distinct rewards to all segments. The inferior results of “Segment as Bandit” demonstrate the benefit of using segment-level rewards throughout PPO training, rather than reverting to a bandit-level reward signal.
> >
> > We have clarified these implementation details in Appendix D of the revised manuscript to improve the transparency and reproducibility of our hybrid baselines.

---

### Review · Reviewer_xEau · 2025-07-18

**Summary Of Contributions:**

This paper presents a segment-level reward modelling method, to improve RLHF for LLMs. Existing RLHF methods either use a bandit formulation, where a reward is assigned to the complete response, or provide token-level rewards. Bandit methods suffer from sparse feedback which is known to make RL difficult, and token-level methods suffer from semantic ambiguity (how much reward should we assign to a sub-word like "ing"?). The proposed takes the middle ground, rewarding segments determined by entropy-based segmentation.

To train the reward model, the authors propose to aggregate segment-level rewards to match existing sequence-level preference labels with a Bradley-Terry loss. During RL (PPO), they use location-aware reward normalization and within-segment reward interpolation, to evenly distribute segment rewards among tokens.

Results show that the model significantly outperforms bandit and token-level approaches on multiple benchmarks (AlpacaEval 2, Arena-hard, MT-Bench) and also outperforms DPO.

**Audience:**

Yes

**Claims And Evidence:**

Yes

**Requested Changes:**

N/A

**Strengths And Weaknesses:**

## Strengths
The paper proposes a natural and simple approach to training and using segment-level reward models in RLHF. I particularly appreciated the extensive set of ablation studies that verify that each component of the proposed approach adds value.

## Weaknesses
The work is motivated by the idea that assigning rewards to individual tokens is problematic, because 1) tokens are not necessarily intuitively meaningful units, and 2) "The contribution of those tokens to the text sequence will inevitably depend on later tokens,
making reward/credit assignment harder". In my opinion this is not very convincing, because "meaningfulness" is quite a vague concept, and the second point also applies to larger chunks such as segments. I am therefore not entirely convinced by the motivation, but the results do show improvements which is ultimately most important.

---

> ### Author Response · Authors · 2025-07-24
> **Response to Reviewer xEau**
>
> We would like to thank Reviewer **xEau** for the positive feedback, especially regarding our ablation studies.
>
> Regarding the motivation for segment-level reward assignment, our goal is to find a more appropriate granularity between bandit (sequence-level) and token-level rewards for effective credit assignment in RLHF. As we show in Figure 2, token-level rewards can be overly fine-grained—sometimes splitting within a single word, which makes the credit assignment oversubtle and potentially noisy. Segment-level actions, determined by entropy-based segmentation, aim to provide a more semantically meaningful unit for reward assignment.
>
> To further support our hypothesis, we have added additional visualizations in Appendix I (Figures 6–8) of the revised manuscript. These figures compare entropy-based segmentation with human-annotated segment boundaries. Most cases show strong alignment, especially at phrase or clause boundaries, suggesting that our method does capture meaningful units in practice. We hope these visualizations further strengthen the motivation and practical value of our approach.

---

### Review · Reviewer_XNou · 2025-07-20

**Summary Of Contributions:**

The paper suggests a modification of the classic bandit formulation of RLHF by proposing a specific instantiation of reward shaping to provide denser rewards in the training of the policy model. Namely, the authors break a sequence into segments based on entropy (with the start of a segment being a token with high conditional entropy) and train a segment level reward model with a specific functional relation to the sequence level pairwise feedback information in the data (i.e., average of the segment-level rewards is the sequence-level reward). Then during policy optimization, the reward for a given segment is evenly distributed across tokens in the segment, after being normalized with a learned mean and standard deviation which take into account the relative location of the segment. Experiments show that this setting outperforms baselines, including those with use coarse grain and fine grained rewards, and ablations show that the particular design choices empirically outperform other simple counterparts.

**Audience:**

Yes

**Claims And Evidence:**

Yes

**Requested Changes:**

1. Could the authors provide more justification for why this proposed approach should outperform RTO? The paper seeks to differentiate itself from other works which utilize token level rewards, but ultimately the proposed approach is another version of assigning token level rewards in a way such that all tokens within a cycle and get the same reward. Intuitively, I would expect that this constraint is actually harmful, because for instance it's not clear why a predictable continuation of a word or phrase should be rewarded the same as the token which begins the word or phrase.
2. Can the authors clarify how they implemented the baselines they compared to in table 1 as well as relative accuracies and losses? Specifically for the sentence baseline, how the sentence level reward model was trained, the relative accuracies of the reward models, and ideally the differences in the number of tokens in a typical sentence versus a segment? For the token baseline, the relative accuracies of the implicit reward model vs. the segment level reward model, the reward normalization, and train loss / reward scores / KL achieved in the evaluated checkpoints?

**Strengths And Weaknesses:**

Strengths:
1. The paper is well-written and easy to follow.
2. Experiments across several different models show the superiority of the approach relative to multiple different baselines.
3. Additional justification, results, and details in the appendix do a nice job of supporting the main paper and giving more context. The comparable observations motivating location-aware normalization, for instance, was interesting.

Weaknesses:
1. Certain design decisions seem rather ad hoc, such as the choice to distribute token rewards within a segment equally and the specific relationship between segment level rewards and the sequence-level reward. It is useful to know that they seem to work alright in practice though.
2. It is not clear why this proposed approach should be better than an approach like RTO that also uses token level rewards, especially if they are well-motivated to approximate actual token level rewards.

---

> ### Author Response · Authors · 2025-07-24
> **Response to Reviewer XNou**
>
> We would like to thank Reviewer **XNou** for the careful review.
> The following are our responses to your questions.
>
>
> > **R1.** Segment Action Space Compared with Token-Level Action space.
>
> Thank you for the question. There is a growing body of work in both reinforcement learning and language modeling demonstrating that assigning rewards at the segment or chunk level provides two key advantages over token- or action-level methods: (1) reward assignment is more semantically meaningful and less noisy, and (2) optimization becomes more efficient, as credit can be assigned to temporally or semantically coherent units rather than isolated tokens or actions. Our own quantitative and visualization results further confirm that segment-level rewards lead to better performance.
>
> Recent RL work [1]  have shown that chunk-level rewards improve both learning efficiency and robustness by better aligning with the underlying structure of meaningful behaviors. This supports the motivation behind our segment-level reward approach for language models.
>
> > **R2.**
> Compared with sentence-level baseline.
>
> For the sentence-level baseline, we segmented text using common punctuation marks such as ".", "!", "?", ";", ",", ":", "\n", and "...". On average, each sentence contains about 15 tokens, whereas our entropy-based segmentation results in shorter segments, averaging around 8.5 tokens.
>
> We evaluated the reward models on RewardBench, where the segment-based reward model achieved an accuracy of 81.81%, and the sentence-level model reached about 81.5%, showing that their accuracies are comparable. However, we found that reward model classification accuracy on such evaluation datasets does not always correspond to downstream PPO policy performance; sometimes, the results can be inconsistent, as also discussed in [2, 3]. This observation is also consistent with recent blog [4], which note that reward model accuracy on evaluation benchmarks like RewardBench or RM-Bench does not always translate to better RLHF policy performance in real-world applications. Therefore, we recommend focusing on our results from actual chat evaluation benchmarks, which better reflect real-world policy performance.
>
> > **R2.**
> Compared with token-level baseline.
>
> We evaluated the reward models on RewardBench, where the segment-based reward model achieved an accuracy of 81.81%, while the token-level model reached about 82.1%. However, as mentioned previously, classification accuracy on RewardBench does not always correspond to downstream task performance. Therefore, we recommend focusing on our results from actual chat evaluation benchmarks for a more reliable comparison.
>
> Regarding policy training, the final policy loss for the segment-based method was 0.1789, compared to 0.3807 for the token-level method. After reward normalization, the average reward of the segment policy model was 0.157, while the average reward of the token policy model was 0.162.
>
> Additionally, in the revised manuscript (see Figure 4), we have included an analysis of the KL divergence between the trained policy and the SFT model in the ablation section. The newly added AlpacaEval win rate vs. KL curves demonstrate that our segment-based PPO policy consistently achieves higher win rates than both the bandit and token-level baselines at the same KL divergence. This shows that our approach makes more effective use of the optimization budget.
>
>
> [1] Li, Qiyang, Zhiyuan Zhou, and Sergey Levine. "Reinforcement Learning with Action Chunking." arXiv preprint arXiv:2507.07969 (2025).
>
> [2] Zheng, Chujie, et al. "Processbench: Identifying process errors in mathematical reasoning." arXiv preprint arXiv:2412.06559 (2024).
>
> [3] Hou, Zhenyu, et al. "Does RLHF Scale? Exploring the Impacts From Data, Model, and Method." arXiv preprint arXiv:2412.06000 (2024).
>
> [4] Wang, Zhilin. "Reward Model Evaluation in June 2025." https://zhilin123.github.io/blog/2025/reward/, 2025.

---

### Decision · Action_Editor_GZtm · 2025-08-24

**Recommendation:** Accept as is

**Additional Comments:**

This manuscript advances reward modeling in RLHF by operationalizing segment-level attribution—a goal which is well-established  in preference optimization (cf. Rafailov et al., 2023; Dong et al., 2023) but rarely achieved in practice. Reviewers commend the degree of empirical validation conducted: added KL–win rate Pareto analyses (reviewer 3Cy9), fine-grained ablations delineating the contribution of segmentation versus policy optimization (reviewers XNou, xEau), and corroboration of segmentation fidelity against human annotations (reviewers VeAx). The method consistently dominates sequence- and token-level baselines as well as DPO across AlpacaEval, Arena-Hard, MT-Bench, and code-generation benchmarks, reflecting alignment with current trends toward modular alignment objectives and compositional generalization (e.g., OpenAI 2024, Anthropic 2024). While some heuristics persist, reviewers converge that the demonstrated robustness and practical tractability render the work both timely and technically substantive. Thus, the work is ready to be published at this time.

**Audience:**

Yes

**Audience Explanation:**

See additional comments.

**Claims And Evidence:**

Yes

**Claims Explanation:**

See additional comments.